# Creating a Public Repository for Joining Private Data

**James Cook**
Independent Researcher*
falsifian@falsifian.org

**Milind Shyani**
Amazon
mshyani@amazon.com

**Nina Mishra**
Amazon
nmishra@amazon.com

## Abstract

How can one publish a dataset with sensitive attributes in a way that both preserves privacy and enables joins with other datasets on those same sensitive attributes? This problem arises in many contexts, e.g., a hospital and an airline may want to jointly determine whether people who take long-haul flights are more likely to catch respiratory infections. If they join their data by a common keyed user identifier such as email address, they can determine the answer, though it breaks privacy. This paper shows how the hospital can generate a private sketch and how the airline can privately join with the hospital's sketch by email address. The proposed solution satisfies pure differential privacy and gives approximate answers to linear queries and optimization problems over those joins. Whereas prior work such as secure function evaluation requires sender/receiver interaction, a distinguishing characteristic of the proposed approach is that it is non-interactive. Consequently, the sketch can be published to a repository for any organization to join with, facilitating data discovery. The accuracy of the method is demonstrated through both theoretical analysis and extensive empirical evidence.

## 1 Introduction

Given a sensitive dataset keyed by individual identities (which must never be revealed), how can we publish a private version of it that can nonetheless be joined by identity to another dataset to produce approximate answers to questions about the join? Can a sender publish this private version to a repository so that receivers can both approximate joint distributions and train ML models, *without any interaction* between the sender and the receiver?

There are *inter-organizational* scenarios where answering these questions is valuable. For example, a healthcare provider may have clinical records keyed by patient email address and a private column such as "has cancer". A company wants to determine if employees who work near toxic waste sites have a higher incidence of cancer. We show how two companies can privately join their data in order to build such predictive models. With the rise of data marketplaces, it is more common for organizations to seek to "decorate" their data by adding new columns from other organizations, while preserving privacy. *Intra-organizational* scenarios also abound. Within a large organization, it is common that data collected in one team is hidden from other teams. With a central repository, not only is data discovery possible, but more private joins can be performed to assess predictive power.

**The Repository Setting.** A stylized view of the problem is shown in Fig. 1. One party, the *sender* S, holds a dataset $D_S$ with two columns: a unique identifier to join on, and a value from a small finite domain (in this case, has cancer encoded as $\pm 1$). S would like to share $D_S$ in a privacy-preserving way with many other organizations without explicitly interacting with each one. To this end, S uploads a private representation of $D_S$ to a public repository. Party R is one of many possible *receivers*, and holds a dataset which has a column with unique identities to join on, and any number of other columns, e.g., latitude and longitude of where their employees work. R's goal is to perform a joint

---

*Work done while employed by Amazon.

37th Conference on Neural Information Processing Systems (NeurIPS 2023).

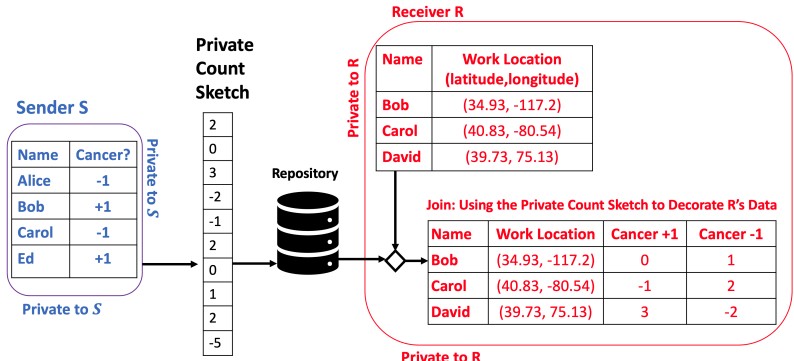

Figure 1: A stylized illustration of the Repository Problem. The sender S uploads a private count sketch capturing people who do and do not have cancer. The receiver R uses the sketch to decorate their data (work location) with a noisy version of S's cancer column. Two noisy columns are generated: one for cancer ($+1$) and another for not ($-1$). R can then build an ML model to predict if employees who work near a toxic waste site are more likely to develop cancer.

computation – red box in Fig. 1. It involves the inner join that would normally include only people in both datasets. But, for privacy, the receiver R does not know about the existence of Alice; and even though David is not in S's data, R still receives a label, which is noise. This paper studies how R can learn functions over the inner join of R and S, e.g., does work location predict cancer positivity.

**Problem statement** Joint computation of linear queries and optimization are described in Subproblems 1.2 and 1.3. The dataset representations should be compatible with both problems simultaneously.

**Problem 1.1.** *Given a dataset $D_S$ with a unique* identity *column and one value column with a small finite domain, publish to a repository a single differentially private representation of $D_S$ which can be combined with any other dataset $D_R$ (so long as $D_R$ has a column of unique identities for joining) to solve Subproblems 1.2 and 1.3.*

A linear query involves applying some function $f$ to each element in the join and outputting the sum.

**Subproblem 1.2** (Linear Queries)**.** *Let $f$ be a real-valued function. Define*

$$\text{SumOverJoin}_{D_R, D_S}(f) = \sum_{\mathit{id} \in \text{ids}(D_R) \cap \text{ids}(D_S)} f(D_R[\mathit{id}], D_S[\mathit{id}])$$

*where $\text{ids}(D)$ is the set of identities that appear in $D$ and $D[\mathit{id}]$ is the value associated with $\mathit{id}$ in $D$. Given access to $D_R$ and a differentially private representation of $D_S$, find an estimate $\hat{\ell}$ so that*

$$\Pr[|\hat{\ell} - \text{SumOverJoin}_{D_R, D_S}(f)| > \mu] < \delta$$

In an optimization query, the goal is to find a function $f$ in a class of functions $\mathcal{F}$ that approximately minimizes $\text{SumOverJoin}_{D_R, D_S}(f)$. Supervised learning with a loss function is a special case.

**Subproblem 1.3** (Optimization Queries)**.** *Given a class $\mathcal{F}$ of real-valued functions, let $f_{\text{opt}} = \arg\min_{f \in \mathcal{F}} \text{SumOverJoin}_{D_R, D_S}(f)$. Given access to $D_R$ and a differentially private representation of $D_S$, find a function $\hat{f}$ such that*

$$\Pr[\text{SumOverJoin}_{D_R, D_S}(\hat{f}) - \text{SumOverJoin}_{D_R, D_S}(f_{\text{opt}}) > \mu] < \delta$$

**Contributions.** Problem 1.1 is solved by building upon work on private count sketches. This paper makes the following contributions. (1) **Method.** We show how to apply the private count sketch to enable a sender to share their data privately, enabling a receiver to approximately join it with their own dataset on individual identifiers to estimate linear queries and learn functions.The identities of the individuals in the sender's data are not revealed. A distinguishing characteristic of the method is that the algorithms are non-interactive, and hence are repository ready. (2) **Analysis.** A non-trivial mathematical analysis in the form of theorems bounding the error. For linear queries, Problem 1.2,

our main result is Thm. 5.1. For optimization queries, Problem 1.3, our main result is Thm. 6.4, proved using a careful combination of tail bounds. (3) **Experiments.** (a) We conduct three sets of experiments on public data to compare our solutions to exact answers obtained by joining the data directly, with no privacy. With a reasonable privacy parameter ($\varepsilon = 1$) we acheive over 92% accuracy on the EMNIST dataset, and applying logistic regression on the UCI Adult dataset, we lose less than 1% accuracy compared to training on the original data with no privacy. (b) We compare to related methods. We adapt the protocol of Zhao et al. [39] to allow for joins and show that our approach is more accurate. We also compare with the LabelDP [21] method on an already-joined dataset, and show that our method performs competitively despite having the harder task of performing a private non-interactive join.

## 2 Related Work

**Privacy.** Differential privacy [19, 18] is a widely used definition that intuitively guarantees that the outcome of a mechanism is essentially unchanged whether an individual is or is not in the dataset. One of the most commonly studied models of privacy in the literature is the *curator model* where a trusted centralized repository holds all personal information. The role of the curator is to answer questions about the data in a differentially private way. In the *function release* problem, the curator must release a single piece of data which can be used to answer an unlimited number of queries [24, 2, 14].

In our case, party S plays the role of curator. The private representation they upload to the public repository can be viewed as an example of function release. It does not depend on the structure of R's dataset or the particular function R aims to compute — S has no knowledge of the parties R who might later download it. Therefore, we can view the sketch as a function accepting as "query" an entire dataset $D_R$ together with the function to apply to the join.

**Private Sketches.** Sketching is a common technique used to summarize a dataset. The technique most relevant to this paper is the Count Sketch [10] which we describe in more detail below. Other influential sketching techniques include count min [16], tug of war [4], locality sensitive hashing [25] and AMS [3]. Private variants of sketching have also been considered wherein, for example, the usual sketch is computed and noise is added. Two examples include the private count sketch [6, 39] and the private count min sketch [39]. Count sketch has been used to solve a wide variety of problems, e.g., estimate histograms, frequencies, quantiles or heavy hitters. We add to this list of uses the ability to support data joins. Other examples of private sketch work include the Johnson-Lindenstrauss transform [7] and the loglog sketch [12] — neither require additional noise for privacy. Private locality sensitive hashing [15] has been considered in the context of private function release.

We are not aware of past work that uses the count sketch to privately join. Nevertheless, it is possible to adapt prior private sketching work to our framework. However, since these solutions were not designed with our application in mind, they do not perform as well – see the experiments in §7.2.

**Local DP.** [6] describes an approach in the local model, where each individual publishes their own noisy data. If individual identities are discarded it precludes joins. If the central server in the local model records individual identities, the identities cannnot be published without compromising privacy. While our techniques are similar to past work in local differential privacy, there are important differences: we apply them in the *repository setting* rather than a local DP setting (see Appendix H for a comparison), the frequencies sketched are always 0 or 1 because unique identifiers are being sketched, and our technique enables joins on those private identifiers. We optimize our use of count sketches for this setting (especially in §6) and provide a novel experimental and theoretical analysis.

**Vertically Partitioned Data.** [17] study a scenario where a user interactively queries multiple independent databases holding vertical partitions of a dataset. Other results include [30, 31, 33, 36]. Note that in our case, the protocol is non-interactive; each database has an identity column which must be used to join the tables; the identities are private; and some identities may be present in one database and not the other.

**Secure Function Evaluation.** In the cryptography community, there is extensive work showing how two parties $A, B$ holding data $x_A, x_B$ can jointly compute $f(x_A, x_B)$ without leaking any information beyond $f(x_A, x_B)$ [37, 23]. For interactive, multi-party computation, cryptographic techniques are a superior solution. However, this paper is on non-interactive solutions for the repository setting. More comparison can be found in Appendix J.

# 3   Preliminaries

**Notation**    $\mathbf{Z}$, $\mathbf{N}$ and $\mathbf{R}$ denote the sets of integers, nonnegative integers and reals. We let $\mathcal{U}$ be some set of unique identifiers for people — e.g. strings representing name and address. For datasets $D, D'$, the *join* $D \bowtie D'$ is defined in Definition 3.1, and $|D| \in \mathbf{N}$ is the number of rows in $D$.

**Datasets**    A *dataset* $D \subseteq \mathcal{U} \times X$ is a set of pairs $(\texttt{id} \in \mathcal{U}, x \in X)$; we call $X$ its *value domain* and its elements *rows*. In our problem there are two datasets: the sender's $(D_S)$ and the receiver's $(D_R)$. The value domain of $D_S$ will always be $X = \{1..k\}$ for some small $k \in \mathbf{N}$, but there is no restriction on the value domain of $D_R$: $X$ could be numbers, text, images, etc. We do not allow any $\texttt{id} \in \mathcal{U}$ to appear more than once in the same dataset, so in particular there are no duplicate rows.

**Joins**    When the same identity appears in two datasets, we want to combine the associated values.

**Definition 3.1.**    The *join* of $D \subseteq \mathcal{U} \times X$ and $D' \subseteq \mathcal{U} \times Y$ is a dataset $D \bowtie D' \subseteq \mathcal{U} \times X \times Y$ defined by

$$(\texttt{id}, x, y) \in D \bowtie D' \iff (\texttt{id}, x) \in D \wedge (\texttt{id}, y) \in D'$$

This an example of a *natural join* [13], using the identity column $D$ and $D'$ have in common. Given a function $f : X \times Y \to \mathbf{R}$, we define the *sum over the join*:

**Definition 3.2.**
$$\mathrm{SumOverJoin}_{D,D'}(f) = \sum_{(\texttt{id},x,y) \in D \bowtie D'} f(x, y)$$

(This is equivalent to the definition in Subproblem 1.2, but written in terms of $D \bowtie D'$.)

**Differential Privacy**    Let $\varepsilon$ and $\delta$ be positive real numbers and $M$ be a randomized algorithm. We say that $M$ is $(\varepsilon, \delta)$-*differentially private* [19] if for all $S \subseteq \mathrm{Range}(M)$ and for all pairs of datasets $D_1$ and $D_2$ that differ in the addition or removal of one row, $\Pr(M(D_1) \in S) \leq e^{\varepsilon} \Pr(M(D_2) \in S) + \delta$, where the probability is taken over the coin flips of $M$. In this work we will always set $\delta = 0$; in this case we say $M$ is $\varepsilon$-differentially private; this is called *pure differential privacy*.

**The two-sided geometric distribution**    The noise we add to ensure privacy will follow the *two-sided geometric distribution* [22] (see notes in Appendix B.1):

**Definition 3.3.**    The *two-sided geometric distribution* with parameter $\alpha$, denoted $\mathrm{TGeom}(\alpha)$, is a distribution over $\mathbf{Z}$ with probability mass function $\Pr[Z = z] = \frac{1-\alpha}{1+\alpha} \alpha^{|z|}$.

**Private Count Sketches**    We use a simplified version of the *count sketch* [10] as a way to encode a dataset $D \subseteq \mathcal{U} \times \{1..k\}$. It is parameterized by a number of buckets $b$ and two hash functions $h : \mathcal{U} \times \{1..k\} \to \{1..b\}, s : \mathcal{U} \times \{1..k\} \to \{-1, +1\}$.

**Definition 3.4.**    Given $b$, $h$ and $s$ as above, the *count sketch* of a dataset $D \subseteq \mathcal{U} \times \{1..k\}$ is a vector $\mathrm{CountSketch}_{b,h,s}(D) = (c_1, \ldots, c_b) \in \mathbf{Z}^b$, where $c_j = \sum_{(\texttt{id},y) \in D : h(\texttt{id},y) = j} s(\texttt{id}, y)$.

Adding noise makes a count sketch differentially private:

**Definition 3.5.**    Given $D$, $b$, $h$ and $s$ as in Definition 3.4, and a privacy parameter $\varepsilon > 0$, the *private count sketch* of $D$ is a random vector $\mathrm{PrivCountSketch}_{b,h,s,\varepsilon}(D) = \mathrm{CountSketch}_{b,h,s}(D) + Z_{\varepsilon} \in \mathbf{Z}^b$ where $Z_{\varepsilon}$ is a vector of $b$ independent samples from $\mathrm{TGeom}(e^{-\varepsilon})$ (Definition 3.3).

Since adding or removing a row from $D$ causes $\mathrm{CountSketch}_{b,h,s}(D)$ to move a distance of 1 in the $\ell_1$ metric, we have [6, 39]:

**Proposition 3.6.**    $\mathrm{PrivCountSketch}_{b,h,s,\varepsilon}(D)$ *is $\varepsilon$-differentially private.*

It is common to repeat the count sketch with multiple independent hash functions to mitigate error caused by hash collisions when there are fewer buckets than items ($b < n$). This is important when the goal is to recover frequencies of repeated items while storing the sketch in as few bits as possible. In contrast, our frequences are all 1 or 0 (present or not), and our goal is to estimate aggregates of many such frequencies. We are not trying to compress a stream, so we take $b \approx n$, at which point

we find the error added by $Z_\varepsilon$ dominates the error from hash collisions. We do not repeat the count sketch because it would improve error only slightly (if at all), and because it would complicate our analysis. In §7.2 we compare to a private sketching method that uses multiple hash functions and Gaussian noise, and find it performs worse than Definition 3.5 for our purposes.

# 4 Method Overview

We begin with intuitive motivation for the method and then a provide an overview of the steps S and R take to solve Problem 1.1. In §§5 and 6 we explain R's computation in detail.

## 4.1 Intuition and alternative approaches

Here is a simplified version of our problem. Suppose parties S and R each have a set of individuals' names $D_S, D_R \subseteq \mathcal{U} = \{A..Z\}^*$. R wishes to estimate the number of names in common: $|D_S \cap D_R|$. The names are sensitive and cannot be shared directly. Instead, S must publish a differentially private representation $C = \text{Private}(D_S)$ without consulting R, and then R must compute their estimate based on $C$ and $D_R$. This is a special case of Subproblem 1.2. Here are some possible approaches.

**Secure function evaluation.** In cryptography, this problem is called *private set intersection cardinality* [20, 26, 32]. Cryptographic solutions are well-studied, but require multiple rounds of interaction. In our *repository setting*, party S publishes a single value $\text{Private}(D_S)$ and may not interact further. See Appendix J for a discussion on this comparison.

**Send the hashes.** One might try choosing a secure hash function $h : \mathcal{U} \to \{0, 1\}^*$ and sending $\text{Private}(D_S) = \{h(x) : x \in D_S\}$. The problem is that R would need to know $h$ to compute $|D_R \cap D_S|$. Knowing $h$ would let them test individuals' membership in $D_S$, compromising privacy.

**Dot product of noisy vectors.** We can view the sets as vectors $D_S, D_R \in \{0, 1\}^{\mathcal{U}}$, with one bit for every possible $\texttt{id} \in \mathcal{U}$. Then $D_S^\top D_R = |D_S \cap D_R|$. Party S can make their vector $\varepsilon$-differentially private by adding noise: $\text{Private}(D_S) = D_S + Z_\varepsilon$, where $Z_\varepsilon \in \mathbf{Z}^{\mathcal{U}}$ is a vector of independent samples from $\text{TGeom}(e^{-\varepsilon})$. Party R can estimate $|D_R \cap D_S| \approx \text{Private}(D_S)^T D_R$. The problem is that $\mathcal{U}$ is too big — if identities are $r$-bit strings, then $\text{Private}(D_S)$ takes more than $2^r$ bits to store.

**Solution: Private count sketch.** The solution we use is a modification of the previous one: S sends a vector with noise added, but first reduces the dimension using a private count sketch.

## 4.2 Protocol

Our solution to Problem 1.1 has two steps: (1) S publishes a sketch and some parameters (Algorithm 1; figure in Appendix A); (2) R downloads the sketch and runs Algorithm 2 (Subproblem 1.2, linear queries) or Algorithm 3 (Subproblem 1.3, choosing the optimal function from a class).

Although the elements of $D_S$ are pairs $(\texttt{id}, y) \in \mathcal{U} \times \{1..k\}$, we treat them as opaque values for the purpose of computing $\text{PrivCountSketch}_{b,h,s,\varepsilon}(D_S)$ — so the domain of the hash functions $h, s$ is $\mathcal{U} \times \{1..k\}$. The time complexity of $\texttt{Sender}$ is $O(|D_S| + b)$, assuming that sampling from $\text{TGeom}(\alpha)$ and computing the hash functions $h$ and $s$ all take constant time.

---

**Algorithm 1** $\texttt{Sender}$

---

Let $\{1..k\}$ be the range of $D_S$'s value column, i.e. $D_S \subseteq \mathcal{U} \times \{1..k\}$.
  1: Choose the privacy budget $\varepsilon$ and the number of hash buckets $b$.
  2: $h, s \leftarrow \texttt{ChooseHashFunctions}(b)$
  3: $C \leftarrow \text{PrivCountSketch}_{b,h,s,\varepsilon}(D_S)$.
  4: Publish $C \in \mathbf{Z}^b$ and the parameters $k, b, h, s, \varepsilon$ to a repository.

---

**Choosing hash functions**  Building a sketch requires first choosing hash functions $h, s$ with an algorithm $\texttt{ChooseHashFunctions}(b)$. This choice has no effect on privacy: even if $\forall x, h(x) = s(x) = 1$, the private count sketch is differentially private. However, higher-quality hash functions will give more accurate results. Thm. 6.4 assumes $h$ and $s$ are sampled uniformly from the set of all functions; in practice, cryptographically secure hash functions should give the same guarantees.

# 5 Linear queries

R can use Algorithm 2 to estimate $\mathrm{SumOverJoin}_{D_R,D_S}(f)$, given access to their own dataset $D_R$ along with $C = \mathrm{PrivCountSketch}_{b,h,s,\varepsilon}(D_S)$ and the associated parameters $k, b, h, s$ sent by S. Its time complexity is $O(k|D_R| + b)$, assuming $f(x,y)$ can be computed in constant time. The computation $Q^{\mathsf{T}}C$ on lines 2–4 can be explained as follows. For every $(\mathtt{id}, x) \in D_R$ and $y \in \{1..k\}$, R multiplies $q(\mathtt{id}, x, y) = s(\mathtt{id}, y) \cdot f(x,y)$ by the entry at index $h(\mathtt{id}, y)$ in S's sketch. In expectation this equals $f(x,y)$ if $(\mathtt{id}, y) \in D_S$ and 0 otherwise. The sum of these values over $(\mathtt{id}, x) \in D_R, y \in \{1..k\}$ is $Q^{\mathsf{T}}C$, and equals $\mathrm{SumOverJoin}_{D_R,D_S}(f)$ in expectation. Here the expectation is taken over the output of `ChooseHashFunctions` and the $\mathrm{TGeom}(\alpha)$ noise.

---

**Algorithm 2** `ReceiverLinearQueries`

---

1: Download $C = \mathrm{PrivCountSketch}_{b,h,s,\varepsilon}(D_S)$ and the parameters $k, b, h, s$ from the repository.
2: Set $q(\mathtt{id}, x, y) \in \mathbf{R}^b$ to be $s(\mathtt{id}, y)f(x,y)$ at entry $h(\mathtt{id}, y)$ and 0 everywhere else.
3: Construct $Q = \sum_{(\mathtt{id},x) \in D_R} \sum_{y=1}^{k} q(\mathtt{id}, x, y)$, the *query vector*.
4: Return $Q^{\mathsf{T}}C$.

---

**Theorem 5.1.** *For any datasets $D_R \subseteq \mathcal{U} \times X, D_S \subseteq \mathcal{U} \times \{1..k\}$, function $f : X \times \{1..K\} \to \mathbf{R}$, privacy parameter $\varepsilon > 0$, number of hash buckets $b$, and accuracy parameter $\mu > 0$, if $h$ and $s$ are drawn from a mutually 4-way independent hash family, the following holds. Let $M$ be an upper bound on $|f(x,y)|$, $k$ be the range of the value column of $D_S$, $n = |D_R \bowtie D_S|, C = \mathrm{PrivCountSketch}_{b,h,s,\varepsilon}(D_S)$. Then*

$$\mathrm{Pr}\left[ \frac{|Q^{\mathsf{T}}C - \mathrm{SumOverJoin}_{D_R,D_S}(f)|}{n} > \mu \right] \leq \frac{M^2|D_R|(k+1)}{\mu^2 n^2} \left( \frac{2e^{-\varepsilon}}{(1-e^{-\varepsilon})^2} + \frac{|D_S|}{b^2} \right)$$

*The probability is taken over $h, s$ and the noise added to $\mathrm{PrivCountSketch}_{b,h,s,\varepsilon}(D_S)$.*

We defer the proof to Appendix C. According to the theorem, the size of the join should be sufficiently large for it to be useful.

# 6 Optimization

Suppose R's dataset has an `id` column and several feature columns, and S's has an `id` column and a label column. In this section we show how R can train a model to predict S's labels from R's features. More generally, if R has a class of functions $\mathcal{F}$ which map $X \times Y \to \mathbf{R}$, we show how R can select $\hat{f} \in \mathcal{F}$ which is close to the optimal function $f_{\mathrm{opt}}$,

$$f_{\mathrm{opt}} := \arg\min_{f \in \mathcal{F}} \mathrm{SumOverJoin}_{D_R,D_S}(f) = \arg\min_{f \in \mathcal{F}} \sum_{(\mathtt{id},x,y) \in D_R \bowtie D_S} f(x,y)$$

in the sense that $|\mathrm{SumOverJoin}_{D_R,D_S}(\hat{f}) - \mathrm{SumOverJoin}_{D_R,D_S}(f_{\mathrm{opt}})|$ is small.

For example, suppose $D_S$ has labels $y \in \{0, 1\}$ and $D_R$ has feature vectors $x \in \mathbf{R}^d$, and R would like to train a logistic regression model on $D_R \bowtie D_S$. Then R can take $\mathcal{F} = \{\ell_\theta : \theta \in \mathbf{R}^d\}$, where $\ell_\theta$ is the logistic loss $\ell_\theta(x,y) = y \log(1 + e^{-\theta^{\mathsf{T}}x}) + (1 - y) \log(1 + e^{\theta^{\mathsf{T}}x})$.

In §§6.1 and 6.2 we show how to compute the estimate $\hat{f}$, assuming we already have a non-private optimization algorithm that accepts *weighted inputs*. (For an alternative without weights, see Appendix E.) In §6.3 we analyze the loss in accuracy due to privacy.

## 6.1 Evaluating one function

Our optimization method is based on a *score* $\Phi_{D_R,C}(f)$ for individual functions $f \in \mathcal{F}$, which R computes based on their dataset $D_R$ and the sketch $C$ sent by S. The score is a proxy for the true objective $\mathrm{SumOverJoin}_{D_R,D_S}(f)$. It is less sensitive to noise and hash collisions than the estimate $Q^{\mathsf{T}}C$ from §5. The cost is that $\Phi_{D_R,C}(f)$ does not actually approximate $\mathrm{SumOverJoin}_{D_R,D_S}(f)$.

Instead, the two have a linear relationship: $\Phi_{D_R,C}(f) \approx \beta \, \mathrm{SumOverJoin}_{D_R,D_S}(f)$ for a positive number $\beta = \beta(D_R, D_S, b, \varepsilon)$ which does not depend on $f$. (We prove this in the appendix in Lemma D.3, as part of the proof of Thm. 6.4.) R does not know $\beta$, but does not need to: the optimization method described in §6.2 boils down to computing $\arg\min_{f \in \mathcal{F}} \Phi_{D_R,C}(f)$.

There are two differences between the score $\Phi_{D_R,C}(f)$ and the linear estimate $Q^\mathsf{T}C$, developed while proving Theorem 6.4. First, we *clip* coordinates of the sketch $C$ to be between $-1$ and $1$.

**Definition 6.1.** For $x \in \mathbf{R}$, $\mathrm{clip}(x)$ is 1 or -1 if $x \geq 1$ or $x \leq -1$; otherwise $\mathrm{clip}(x) = x$. The *clipped version* of a vector $C \in \mathbf{R}^b$ is a vector with clipped entries: $\mathrm{clip}(C)_i = \mathrm{clip}(C_i)$.

(Since our sketches have integer coordinates, $\mathrm{clip}(x)$ equals the sign $\mathrm{sgn}(x) \in \{-1, 0, 1\}$. In §7.2 we try a sketch with non-integer coordinates.) Second, when $h$ hashes more than one row $(\mathtt{id}, x) \in D_R$ to the same coordinate of $C$, we reweight the terms corresponding to those rows.

**Definition 6.2.** The *score* of $f \in \mathcal{F}$ is defined as follows. Let $N_R(a)$ be the number of possible pairs $(\mathtt{id}, y)$ where $\mathtt{id}$ appears in $D_R$, $y$ is any value in $\{1..k\}$, and $h(\mathtt{id}, y) = a$:

$$\Phi_{D_R,C}(f) := \sum_{(\mathtt{id},x) \in D_R} \sum_{y=1}^{k} \frac{s(\mathtt{id}, y) \, \mathrm{clip}(C)_{h(\mathtt{id},y)} f(x, y)}{N_R(h(\mathtt{id}, y))}$$

## 6.2 Performing the optimization

Here we show how R can compute the estimate $\hat{f} \in \mathcal{F}$, given their own dataset $D_R$, the parameters $k, b, h, s, \varepsilon$, and $C = \mathrm{PrivCountSketch}_{b,h,s,\varepsilon}(D_S)$ downloaded from the repository. In short, our method is simply to find the function $f \in \mathcal{F}$ that minimizes the score defined in the previous section:

**Definition 6.3.** The *estimated optimal function in $\mathcal{F}$* is $\hat{f} := \arg\min_{f \in \mathcal{F}} \Phi_{D_R,C}(f)$.

If $\mathcal{F}$ is a small finite set, this definition directly leads to an algorithm: R computes $\Phi_{D_R,C}(f)$ for every $f \in \mathcal{F}$ and chooses the $\arg\min$ directly. For larger $\mathcal{F}$s we will assume R has an algorithm WeightedOpt to find the optimal $f$ on a *weighted dataset*. A weighted dataset $\tilde{D}$ is a collection of $(w, x, y)$ triples where $w \in \mathbb{R}$ is called the *weight* and $(x, y)$ is a valid input to functions in $\mathcal{F}$. To be precise, WeightedOpt minimizes

$$\mathrm{SumOverDataset}_{\tilde{D}}(f) := \sum_{(w_{\mathtt{id},y}, x, y) \in \tilde{D}} w_{\mathtt{id},y} f(x, y)$$

Many algorithms for training machine learning models admit a weight associated with each training example; such an algorithm could play the role of WeightedOpt. R can now find the optimal function by implementing Algorithm 3. The time complexity of this algorithm is $O(k|D_R|) + W(k|D_R|)$, where $W(n)$ is the run-time complexity of WeightedOpt on a dataset with $n$ elements.

By Definitions 6.2 and 6.3, the output of WeightedOpt is identical to the estimated optimal function $\hat{f}$. The question of whether $\hat{f}$ is any good is addressed in §6.3.

---

**Algorithm 3** ReceiverOptimize

---

1: Download $C = \mathrm{PrivCountSketch}_{b,h,s,\varepsilon}(D_S)$ and the parameters $k, b, h, s$.
2: $w_{\mathtt{id},y} \leftarrow s(\mathtt{id}, y) \, \mathrm{clip}(C_{h(\mathtt{id},y)}) / N_R(h(\mathtt{id}, y))$ for $y = 1, \ldots, k$
3: Construct $\tilde{D} = \tilde{D}(b, h, s, D_R, C)$ by creating $(w_{\mathtt{id},y}, x, y)$ for each element $(\mathtt{id}, x) \in D_R$ and $y \in \{1..k\}$.          $//note : |\tilde{D}| = k|D_R|$
4: Return WeightedOpt$(\tilde{D}, \mathcal{F})$

---

## 6.3 Analysis

Here we analyze the accuracy of our method in the context of learning a classifier. We state and prove a more general result in Appendix D, and prove the below results in Appendices D.2 and D.3.

We bound the classification error of a model trained using our method on features from $D_R$ and binary labels from $D_S$ (so $k = 2$). Let $\mathcal{F}$ be a set of classifiers $f : X \to \{0, 1\}$, and for $f \in \mathcal{F}$ let

$L_f \in \mathbf{N}$ be the number of classifications errors on $D_R \bowtie D_S$. To apply Algorithm 3, we convert this to an equivalent problem: let $\mathcal{G}$ be the set of corresponding error functions $e_f(x,y) = |f(x) - y|$ for $f \in \mathcal{F}$, so $L_f = \text{SumOverJoin}_{D_R, D_S}(e_f)$. Then we can ask Algorithm 3 to find the best $e_f \in \mathcal{G}$, which corresponds to the best classifier $f$.

**Theorem 6.4.** *For any datasets $D_R \subseteq \mathcal{U} \times X$, $D_S \times \mathcal{U} \times \{1..k\}$, function class $\mathcal{F}$, privacy parameter $\varepsilon > 0$, number of hash buckets $b$ and accuracy parameter $\mu > 0$, if $h$ and $s$ are drawn uniformly at random from the set of all[2] functions $\mathcal{U} \times \{1..k\} \to \{1..b\}$, the following holds. Let $f_{\text{opt}}$ be the classifier that minimizes $L_f$, and let $\hat{f}$ be the classifier corresponding to the function $e_{\hat{f}}$ output by Algorithm 3. Let $d$ be the VC dimension of $\mathcal{F}$, $n = |D_R \bowtie D_S|$, and*

$$\phi = \frac{|D_R|}{n} \quad \psi = \frac{|D_S|}{n} - 1 \quad W_R = \min\left\{1, \frac{b}{5k|D_R|}\right\} \quad W_S = \min\left\{\frac{1}{4}, \varepsilon, \frac{1}{14}\sqrt{\frac{b}{|D_S|}}\right\}$$

*Then*

$$\Pr\left[\frac{L_{\hat{f}} - L_{f_{\text{opt}}}}{n} > \mu\right] \leq \exp\left(-\Omega\left(\frac{W_R W_S}{\phi + \psi}\mu^2 n - d \log n\right)\right)$$

For reference, we define VC dimension in Appendix B.4. Adding assumptions can simplify the bound. This corollary highlights the dependence on the privacy parameter $\varepsilon$:

**Corollary 6.5.** *Assume $D_R$ and $D_S$ have the same set of identities, so the join is "perfect" in the sense that $|D_R| = |D_S| = |D_R \bowtie D_S|$. Let $n = |D_R \bowtie D_S|$ and let $d$ be the VC dimension of $\mathcal{F}$. Assume also that $b = \Omega(n)$. Let $f_{\text{opt}}$ be the classifier that minimizes $L_f$, and let $\hat{f}$ be the classifier corresponding to the function $e_{\hat{f}}$ output by Algorithm 3. Then for any $\mu > 0$,*

$$\Pr\left[\frac{L_{\hat{f}} - L_{f_{\text{opt}}}}{n} > \mu\right] \leq \exp(-\Omega(\mu^2 n \varepsilon - d \log n))$$

# 7 Experiments

The goal of the experiments is to evaluate the accuracy of our solutions and quantify the effect of noise and join size on downstream tasks. To simulate parties S and R, for each dataset, we add a unique id to each row, then split its columns into $D_R$ with all of the features and $D_S$ with labels. In §7.1 we learn a function that, given features $x$, predicts labels $y \in \{1..k\}$ using Algorithm 3 and a private count sketch. In Appendix F we estimate a joint distribution using Algorithm 2. In both cases, we find that the test error drops as we increase $\varepsilon$ (meaning the privacy requirement is relaxed) or the sketch dimension, or as the join size increases ($D_R, D_S$ have more ids in common). In §7.2 we compare our results with the private linear queries protocol of Zhao et al. [39]. We also conduct experiments with multi-way joins and with non-unique identifiers in Appendix I.

**Datasets.** Two datasets are used in the experiments. (1) UC Irvine Adult dataset [1]. We predict if income is greater than 50K based on categorical features such as *age*, *workclass* and *education*. (2) Extended MNIST, for which we predict labels based on images of handwritten digits.

## 7.1 Optimization: Prediction Findings

We evaluate our method with two machine learning tasks: logistic regression on the UCI Adult dataset, and a neural net for Extended MNIST. We capped the cross-entropy loss by replacing predicted probabilities $\hat{y}$ with $\max\{\hat{y}, 10^{-4}\}$. This is necessary since Algorithm 3 can produce negative weights, which would lead to divergence in the optimization if loss were not bounded.

In each case, we conduct several experiments varying the privacy parameter[3] $\varepsilon$ and the sketch dimension $b$. All accuracy numbers are measured on the test set directly, with no simulated join. The right halves of Figs. 2 and 3 show the results, with different lines for different sketch dimensions $b$. A dashed line shows the test accuracy of a model trained directly on the training set, without privacy.

---

[2]In practice, cryptographically secure hash functions should give the same guarantees.

[3]Ponomareva et al. [29, §5.2.1] suggest $\varepsilon \leq 1$ gives a "strong" privacy guarantee and $\varepsilon < 10$ is "reasonable".

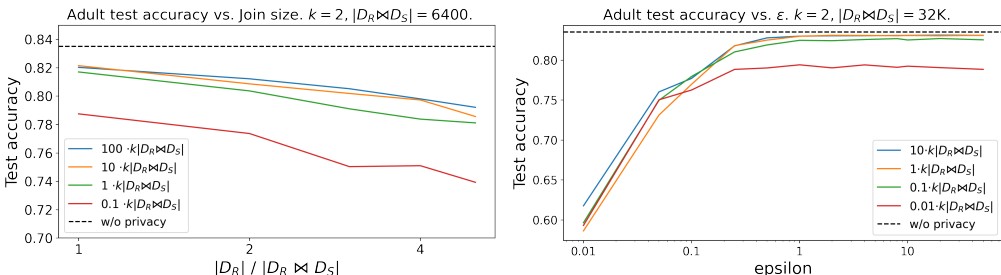

Figure 2: Test accuracy of logistic regression on the UCI Adult salary prediction task (§7.1).

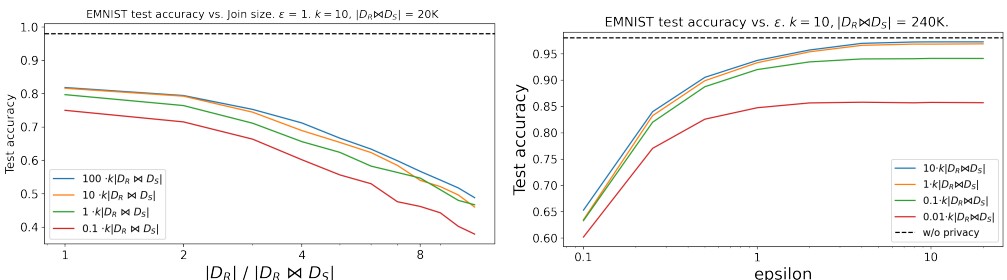

Figure 3: Test accuracy of a deep neural net on the Extended MNIST dataset (§7.1).

Imperfect joins, where $D_R$ has some IDs not present in $D_S$, are simulated by randomly removing most of the rows in the training set, and then adding some back only to $D_R$. The left halves of Figs. 2 and 3 show the results. The performance drops as this fraction increases, which is expected since unmatched rows in $D_R$ are assigned random labels that add noise to the training. In practice, party R would benefit by filtering $D_R$ to contain only identities likely to be in $D_S$, e.g., if $D_S$ contains people in Ohio, R should also filter $D_R$ to Ohio. Note that on the right panel of Fig. 3, when $\varepsilon = 1$, the test accuracy is around 92%, however, on the left panel of Fig. 3, the performance drops to 80% for $\varepsilon = 1$. This is due to the smaller training set size (240K on the right vs 20K on the left).

We also train a neural net on the EMNIST `bymerge` dataset consisting of 760K images of handwritten digits and uppercase and lowercase letters, with 47 classes. We conduct this experiment to investigate the effects of large label domain sizes, i.e., large $k$. Fig. 4, shows test accuracy as a function of the number of labels. For each run, we randomly chose out of 47 classes, and applied our method with $\varepsilon = 1$. The figure shows that, as expected, the performance degrades significantly as $k$ increases, but the method is still viable with $k = 45$. Note that in this experiment, the size of the dataset, and thus the join size, changes as we change $k$.

Our results are consistent with the dependence of Thm. D.2 on the parameters $\varepsilon$, $b$, and $\phi = |D_R|/|D_R \bowtie D_S|$: test accuracy improves as $\varepsilon$ or $b$ increase, or $|D_R|/|D_R \bowtie D_S|$ or $k$ decrease. Training details can be found in Appendix G.

### 7.2 Comparison with other methods

To our knowledge, there is no past work on non-interactive private joins in the repository setting. However, one can replace our simplified private count sketch (Definition 3.5) with other private sketches. To this end, we incorporate Zhao et al's [39] private linear sketches into our method. Party S generates a sketch of $(\mathtt{id}, \mathrm{value})$ pairs as before, but builds the sketch using Algorithms 1 and 3 from Zhao et al. instead of our Definition 3.5. Then, line 2 of Algorithm 3 is replaced with LinearSketchQuery from Zhao et al.s' Algorithm 2: $w_{\mathtt{id},y} \leftarrow \mathrm{clip}(\mathrm{LinearSketchQuery}(\mathtt{id}, y))$.

The EMNIST experiment is run with different combinations of Zhao et al.'s parameters $\gamma, \beta, \rho$. Their protocol satisfies $\rho$-zCDP; we use Proposition 1.4 of [8] to compare our results by equating $\rho = \frac{1}{2}\varepsilon^2$. The number of hash buckets $b$ is related to $\gamma$ as $b = 1/\gamma$ and the number of hash functions (one for us) is given by $\log(2/\beta)$.

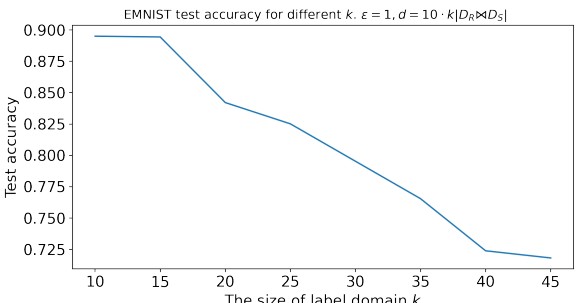

Figure 4: Test accuracy as the domain size of the labels, $k$, increases.

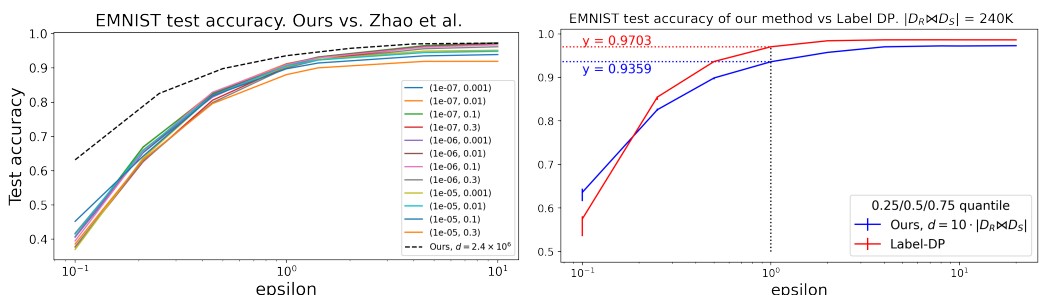

Figure 5: Comparisons with other baselines. The definition of neighboring datasets is different for LabelDP and ours: $2\varepsilon_{\text{ours}} \approx \varepsilon_{\text{theirs}}$, roughly. We have taken this into account in the right panel.

Zhao et al. describe two sketch implementations, Count-Min and CountSketch. Count-Min sketch is a random classifier on EMNIST for $\varepsilon \lesssim 10$, possibly because Count-Min is designed to avoid underestimating frequencies, whereas in our case it is more important to be correct on average. CountSketch leads to better results, Fig. 5. For $\beta = 0.3$, there is one hash function, as with our method, and the remaining differences between the two methods is the kind of added noise (Gaussian vs two-sided geometric), and that LinearSketchQuery omits the denominator $N(h(\texttt{id}, y))$ on Line 2 of our Algorithm 3. Our method provides better results, especially for smaller $\varepsilon$.

The accuracy of the model is roughly proportional to the number of correct vs. incorrect labels reconstructed in the join. For the corresponding values of $\varepsilon$ and $(\rho, \beta, \gamma)$ in the two methods, the number of incorrect labels reconstructed is fewer for the two sided geometric noise than Gaussian. Gaussian noise has thinner tails which is helpful for reducing the probability of large error values but not helpful in our application where queries are in any case clipped to the range $[-1, 1]$.

We also compare with LabelDP [21], which is a Local DP protocol that protects the *labels*. LabelDP is not designed to perform joins, and so we run it without splitting the training set. Results are shown in the right panel of Fig. 5. The definition of "neighbouring datasets" differs in the two settings: we use the "add/remove" definition where a single row is added or removed, but in LabelDP's definition, a single value is changed. To account for this, we double the privacy budget for LabelDP, so for example at $\varepsilon = 1$ on the $x$-axis, we actually give LabelDP a privacy budget of $\varepsilon = 2$.

The LabelDP method benefits from not needing to perform a join, and so we expect it to perform better than our method, where the parties must join using a non-interactive protocol. We were surprised to see our method perform better for small $\varepsilon$, and do not understand why this happens.

**Future Work.** This work focused on when the sender has a single private label column. To broaden applicability, data with multiple columns and data of different types, such as real-valued or text, deserve consideration. New privacy challenges arise in the streaming setting where organizations continuously receive new data and discard outdated data.

**Acknowledgments**

We would like to thank Joan Feigenbaum, Bill Horne, Yonatan Naamad, Aaron Roth, Doug Terry, Tal Wagner, and the anonymous reviewers for helpful feedback, comments and suggestions that improved this paper.

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

# Appendices

## A   Illustration of Algorithm 1

Figure 6 provides a visual depiction of the private count sketch that the sender uploads to the public repository. The noise values $Z_\varepsilon$ are sampled from $\mathrm{TGeom}(e^{-\varepsilon})$ (Def. 3.3). For example, if we choose $\varepsilon = 1$ (which gives a "strong" privacy guarantee [29, §5.2.1]), then the variance of each $Z_\varepsilon$ is about 1.84.

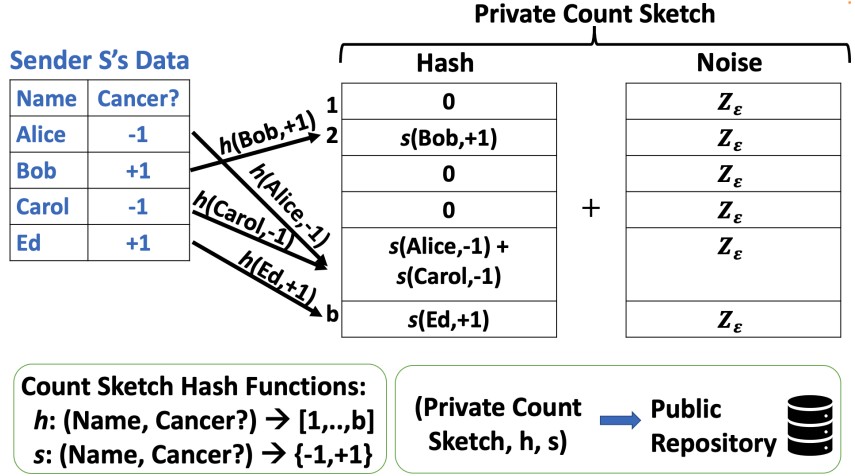

Figure 6:   Visual depiction of Algorithm 1

## B   Some Useful Definitions and Theorems

### B.1   More on the two-sided geometric distribution

We repeat the definition for convenience:

**Definition 3.3.**  The *two-sided geometric distribution* with parameter $\alpha$, denoted $\mathrm{TGeom}(\alpha)$, is a distribution over $\mathbf{Z}$ with probability mass function $\Pr[Z = z] = \frac{1-\alpha}{1+\alpha}\alpha^{|z|}$.

$\mathrm{TGeom}(\alpha)$ is the same distribution as the difference between two independent geometric random variables, $\mathrm{Geom}(p = 1 - \alpha)$. This fact can be verified directly from the probability mass functions, and it provides a convenient way to sample from the distribution when using a library that provides the geometric distribution. From this it immediately follows that the expectation and variance of $\mathrm{TGeom}(\alpha)$ are given by:

**Lemma B.1.**  *For $Z \sim \mathrm{TGeom}(\alpha)$, $\mathbf{E}[Z] = 0$ and $\mathrm{Var}[Z] = \frac{2\alpha}{(1-\alpha)^2}$.*

### B.2   Mutually independent hash families

**Definition B.2.**  A distribution $H$ over pairs of hash functions $h : U \to \{1..b\}, s : U \to \{-1, 1\}$ is a *mutually $k$-way independent hash family* iff for any fixed distinct inputs $x_1, \ldots, x_k$ and outputs $y_1, \ldots, y_k \in \{1..b\}$ and $z_1, \ldots, z_k \in \{-1, 1\}$,

$$\Pr[\forall i \in \{1..k\}\ h(x_i) = y_i \wedge s(x_i) = z_i] = \left(\frac{1}{2b}\right)^k$$

In other words, the outputs $h(x_1), \ldots, h(x_k), s(x_1), \ldots, s(x_k)$ are distributed uniformly over $\{1..b\}^k \times \{-1, 1\}^k$.

### B.3 McDiarmid's Inequality

**Theorem B.3** (McDiarmid's Inequality). *Let $f : \mathcal{X}_1 \times \cdots \times \mathcal{X}_n \to \mathbf{R}$ have the property that for every index $i \in \{1..n\}$ and every pair of sequences $\vec{x}, \vec{y} \in \mathcal{X}_1 \times \cdots \times \mathcal{X}_n$ that differ only in the $i$-th coordinate, $|f(\vec{x}) - f(\vec{y})| \leq c_i$. Then if $X_1 \sim \mathcal{X}_1, \ldots, X_n \sim \mathcal{X}_n$ are independent random variables, then for any $\mu \geq 0$,*

$$\Pr[|f(X_1, \ldots, X_n) - \mathbf{E}[f(X_1, \ldots, X_n)]| \geq \mu] \leq 2 \exp\left(-\frac{2\mu^2}{\sum_{i=1}^n c_i}\right)$$

### B.4 Measuring the complexity of a function class

When analyzing an optimization algorithm, it is useful to have a measure of the *complexity* of a class of functions. One such measure is the *uniform covering number*.

**Definition B.4** (Covering number in $\mathbf{R}^n$ [5]). An $\ell_\infty$ $\mu$-*cover* of a set $S \subseteq \mathbf{R}^n$ is a set $C \subseteq S$ such that $\forall x \in S \ \exists c \in C \ |x - c|_\infty < \mu$. The $\ell_\infty$ $\mu$-*covering number* of $S$, denoted $\mathcal{N}(\mu, S, \ell_\infty)$, is the cardinality of the smallest such $C$.

**Definition B.5** (Uniform covering number of a function class [5]). Given a function $f : X \to \mathbf{R}$ and a sequence $\vec{x} = (x_1, \ldots, x_n) \in X^n$, the *restriction of $f$ to $\vec{x}$* is

$$f|_{\vec{x}} = (f(x_1), \ldots, f(x_n)) \in \mathbf{R}^n$$

The restriction of a class of functions $\mathcal{F}$ is the set of restrictions of the functions in it:

$$\mathcal{F}|_{\vec{x}} = \{f|_{\vec{x}} : f \in \mathcal{F}\}$$

The *uniform covering number* $\mathcal{N}_\infty(\mu, \mathcal{F}, n)$ is the largest covering number of any restriction $\mathcal{F}|_{\vec{x}}$ to a length-$n$ sequence:

$$\mathcal{N}_\infty(\mu, \mathcal{F}, n) = \max_{\vec{x} \in X^n} \mathcal{N}(\mu, \mathcal{F}|_{\vec{x}}, \ell_\infty)$$

The *Vapnik-Chervonenkis dimension* or VC-dimension is another measure of complexity that applies to boolean-valued functions. In the case of boolean-valued functions, the uniform covering number is called the *growth function*:

**Definition B.6** (Growth function [5]). Let $\mathcal{F}$ be a class of functions $f : X \to \{0, 1\}$. The *growth function* of $\mathcal{F}$ is a function $\Pi_\mathcal{F} : \mathbf{N} \to \mathbf{N}$ defined by $\Pi_\mathcal{F}(m) = \mathcal{N}_\infty(\frac{1}{2}, \mathcal{F}, m)$.

($\frac{1}{2}$ could be replaced by any number less than 1.) Note that $\Pi_\mathcal{F}(m) \leq 2^m$ since there are only $2^m$ possible restrictions $f|_{\vec{x}}$ for a length-$m$ sequence $\vec{x}$.

**Definition B.7** (VC Dimension [5]). Let $\mathcal{F}$ be a class of functions $f : X \to \{0, 1\}$. The *VC dimension* of $\mathcal{F}$, denoted $\text{VCdim}(\mathcal{F})$, is the largest $d$ such that $\Pi_\mathcal{F}(d) = 2^d$.

The VC dimension summarizes the behaviour of the growth function in a way captured by the following lemma:

**Lemma B.8** (Theorem 3.7 of [5]). *Let $\mathcal{F}$ be a class of boolean-valued functions with VC dimension $d$. Then for any $m \geq 1$, $\Pi_\mathcal{F}(m) \leq m^d + 1$.*

We will make use of a simple connection between the complexity of a collection of classifiers $f$ and of the corresponding set of error functions $e_f$:

**Lemma B.9.** *Let $\mathcal{F}$ be a class of functions $f : X \to \{0, 1\}$. For $f \in \mathcal{F}$ define $e_f : X \times \{0, 1\} \to \{0, 1\}$ by $e_f(x, y) = |f(x) - y|$, and let $\mathcal{G} = \{e_f : f \in \mathcal{F}\}$. Then*

$$\text{VCdim}(\mathcal{G}) = \text{VCdim}(\mathcal{F})$$

*Proof.* Let $d_\mathcal{G} = \text{VCdim}(\mathcal{G})$ and $d_\mathcal{F} = \text{VCdim}(\mathcal{F})$.

First we'll show $d_\mathcal{G} \geq \text{VCdim}(\mathcal{G})$. Unpacking the definitions of VC dimension and growth function, we see there is $\vec{x} = (x_1, \ldots, x_{d_\mathcal{F}}) \in X^{d_\mathcal{F}}$ such that $\mathcal{F}|_{\vec{x}}$ includes every one of the $2^{d_\mathcal{F}}$ sequences in $\{0, 1\}^{d_\mathcal{F}}$. Let $\vec{z} = ((x_1, 0), \ldots, (x_{d_\mathcal{F}}, 0))$. Observe that for any $f \in \mathcal{F}$, $f|_{\vec{x}} = e_f|_{\vec{z}} \in \{0, 1\}^{d_\mathcal{F}}$. Therefore $\mathcal{G}_{\vec{z}}$ includes all $2^{d_\mathcal{F}}$ sequences, and so $\text{VCdim}(\mathcal{G}) \geq d_\mathcal{F}$.

To show $d_\mathcal{G} \geq \mathrm{VCdim}(\mathcal{G})$, note there is some $\vec{z} = ((x_1, y_1), \ldots, (x_{d_\mathcal{G}}, y_{d_\mathcal{G}})) \in (X \times \{0,1\})^{d_\mathcal{G}}$ such that $\mathcal{G}|_{\vec{z}}$ includes every sequence in $\{0,1\}^{d_\mathcal{G}}$. Let $\vec{x} = (x_1, \ldots, x_{d_\mathcal{G}}) \in X^{d_\mathcal{G}}$. Given a sequence $s \in \{0,1\}^{x_{d_\mathcal{G}}}$, we would like to show $s \in \mathcal{F}|_{\vec{x}}$. Indeed, define the sequence $s' \in \{0,1\}^{x_{d_\mathcal{G}}}$ by $s_i' = |s_i - y_i|$, and let $e_f$ be a function in $\mathcal{G}$ (corresponding to some $f \in \mathcal{F}$ such that $e_f|_{\vec{z}} = s'$. Then for any $i \in \{1..d_\mathcal{G}\}$, $f(x_i) = |e_f(x_i) - y_i| = |s_i' - y_i| = s_i$, and so $f|_{\vec{x}} = s$. We were able to do this for an arbitrary $s$, so $\mathcal{F}_{\vec{x}}$ includes all $2^{d_\mathcal{G}}$ sequences, so $\mathrm{VCdim}(\mathcal{F}) \geq d_\mathcal{G}$. $\qquad\square$

## C  Linear Queries: Proof of Theorem 5.1

We prove Thm. 5.1, repeated here for convenience:

**Theorem 5.1.** *For any datasets $D_R \subseteq \mathcal{U} \times X, D_S \subseteq \mathcal{U} \times \{1..k\}$, function $f : X \times \{1..K\} \to \mathbf{R}$, privacy parameter $\varepsilon > 0$, number of hash buckets $b$, and accuracy parameter $\mu > 0$, if $h$ and $s$ are drawn from a mutually 4-way independent hash family, the following holds. Let $M$ be an upper bound on $|f(x, y)|$, $k$ be the range of the value column of $D_S$, $n = |D_R \bowtie D_S|, C = \mathrm{PrivCountSketch}_{b,h,s,\varepsilon}(D_S)$. Then*

$$\Pr\left[\frac{|Q^\mathsf{T}C - \mathrm{SumOverJoin}_{D_R,D_S}(f)|}{n} > \mu\right] \leq \frac{M^2|D_R|(k+1)}{\mu^2 n^2}\left(\frac{2e^{-\varepsilon}}{(1-e^{-\varepsilon})^2} + \frac{|D_S|}{b^2}\right)$$

*The probability is taken over $h, s$ and the noise added to $\mathrm{PrivCountSketch}_{b,h,s,\varepsilon}(D_S)$.*

The proof follows by applying Chebyshev's inequality to the following lemma:

**Lemma C.1.** *If $h$ and $s$ are drawn from a mutually 4-way independent hash family (Definition B.2), then*

$$\mathbf{E}[Q^\mathsf{T}C] = \mathrm{SumOverJoin}_{D_R,D_S}(f)$$

*and*

$$\mathrm{Var}[Q^\mathsf{T}C] \leq |f|_\infty^2 |D_R|(k+1)\left(\frac{2e^{-\varepsilon}}{(1-e^{-\varepsilon})^2} + \frac{|D_S|}{b^2}\right)$$

*where $k$ is the number of possible values allowed in $S$'s value column, $|f|_\infty = \sup_{x,y} f(x,y)$, $C = \mathrm{PrivCountSketch}_{b,h,s,\varepsilon}(D_S)$, and the randomness is over the choice of $h$ and $s$ and the noise from $\mathrm{TGeom}(e^{-\varepsilon})$ that was added to $C$.*

*Proof.* We write $C = Z + \sum_{(\mathtt{id},y)\in D_S} C(\mathtt{id}, y)$ where $C(\mathtt{id}, y)$ is a vector with $s(\mathtt{id}, y)$ at coordinate $h(\mathtt{id}, y)$ and 0 everywhere else, and $Z$ is a vector of independent $\mathrm{TGeom}(e^{-\varepsilon})$ random variables.

For every $\mathtt{id}, x, y$, we have

$$q(\mathtt{id}, x, y)^\mathsf{T}C(\mathtt{id}, y) = f(x, y) \tag{1}$$

On the other hand, if $(\mathtt{id}, y') \neq (\mathtt{id}', y)$ then $\mathbf{E}[q(\mathtt{id}, x, y')^\mathsf{T}C(\mathtt{id}', y)] = 0$. This follows from the facts that $s$ is independent of $h$ and that $s$ is two-way independent and so $\mathbf{E}[s(\mathtt{id}, y')s(\mathtt{id}', y)] = 0$. Finally, $\mathbf{E}[q(\mathtt{id}, x, y)^\mathsf{T}Z]$ is zero, since $Z$ is independent of $Q$ and has expectation zero. So:

$$\mathbf{E}[Q^\mathsf{T}C] = \mathbf{E}\left[\left(\sum_{(\mathtt{id},x)\in D_R}\sum_{y'=1}^{k} q(\mathtt{id}, x, y')\right)^\mathsf{T} \cdot \left(Z + \sum_{(\mathtt{id}',y)\in D_S} C(\mathtt{id}', y)\right)\right]$$

$$= \sum_{(\mathtt{id},x)\in D_R}\sum_{y'=1}^{k}\sum_{(\mathtt{id}',y)\in D_S} \mathbf{E}[q(\mathtt{id}, x, y')^\mathsf{T}C(\mathtt{id}', y)]$$

$$= \sum_{(\mathtt{id},x,y)\in D_R\bowtie D_S} f(x, y)$$

$$= \mathrm{SumOverJoin}_{D_R,D_S}(f)$$

Moving on to the variance, we have

$$\mathrm{Var}[Q^\mathsf{T}C] = \mathbf{E}[(Q^\mathsf{T}C - \mathrm{SumOverJoin}_{D_R,D_S}(f))^2]$$

$$= \mathbf{E}[(A + B - \mathrm{SumOverJoin}_{D_R,D_S}(f))^2]$$

where $A = \sum_{(\texttt{id},x) \in D_R} \sum_{y=1}^{k} q(\texttt{id}, x, y)^\mathsf{T} Z$ and

$$B = \sum_{(\texttt{id},x) \in D_R} \sum_{y'=1}^{k} \sum_{(\texttt{id}',y) \in D_S} q(\texttt{id}, x, y')^\mathsf{T} C(\texttt{id}', y)$$

Using (1) can write

$$B - \text{SumOverJoin}_{D_R, D_S}(f) = \sum_{(\texttt{id},x) \in D_R} \sum_{y'=1}^{k} \sum_{(\texttt{id}',y) \in D_S} [(\texttt{id}, y') \neq (\texttt{id}', y)] q(\texttt{id}, x, y')^\mathsf{T} C(\texttt{id}', y)$$

$$= \sum_{(\texttt{id},x) \in D_R} \sum_{y'=1}^{k} \sum_{(\texttt{id}',y) \in D_S}$$

$$[(\texttt{id}, y') \neq (\texttt{id}', y)][h(\texttt{id}, y') = h(\texttt{id}', y)] \cdot$$

$$s(\texttt{id}, y') s(\texttt{id}', y) f(x, y) \tag{2}$$

where $[(\texttt{id}, y') \neq (\texttt{id}', y)]$ denotes a function which is 0 whenever $\texttt{id} = \texttt{id}'$ and $y' = y$ and 1 otherwise, and similarly $[h(\texttt{id}, y') = h(\texttt{id}', y)]$ is 1 when the hashes are equal. We write,

$$\text{Var}[Q^\mathsf{T} C] = \mathbf{E}[A^2] + 2\,\mathbf{E}[A(B - \text{SumOverJoin}_{D_R, D_S}(f))] + \mathbf{E}[(B - \text{SumOverJoin}_{D_R, D_S}(f))^2]$$

and compute each term separately.

$$\mathbf{E}[A^2] = \sum_{(\texttt{id},x) \in D_R} \sum_{y=1}^{k} f(x, y)^2 \frac{2e^{-\varepsilon}}{(1 - e^{-\varepsilon})^2} \leq \frac{2k|D_R|e^{-\varepsilon}|f|_\infty^2}{(1 - e^{-\varepsilon})^2} \tag{3}$$

Note that cross terms in $A^2$ are zero in expectation because each includes a product $s(\texttt{id}, y) s(\texttt{id}', y')$ where $(\texttt{id}, y) \neq (\texttt{id}', y')$. The variance of the two-sided geometric distribution comes from Lemma B.1. We have

$$2\,\mathbf{E}[A(B - \text{SumOverJoin}_{D_R, D_S}(f))] = 0 \tag{4}$$

since (referring to (2)) each term includes a product of distinct ID-y pairs $s(\texttt{id}, y) s(\texttt{id}', y') s(\texttt{id}'', y'')$ or $s(\texttt{id}, y) s(\texttt{id}', y')^2$ which are both 0 in expectation since $s$ is 3-way independent. Finally, looking at (2), we see

$$\mathbf{E}[(B - \text{SumOverJoin}_{D_R, D_S}(f))^2] = \sum_{\substack{(\texttt{id},x) \in D_R \\ y' \in \{1..k\} \\ (\texttt{id}',y) \in D_S \\ (\texttt{id}'',x') \in D_R \\ y''' \in \{1..k\} \\ (\texttt{id}''',y'') \in D_S}} C \cdot D \cdot f(x, y) \cdot f(x', y'')$$

where $C = [(\texttt{id}, y') \neq (\texttt{id}', y)][(\texttt{id}'', y''') \neq (\texttt{id}''', y'')] \Pr[h(\texttt{id}, y') = h(\texttt{id}', y) \wedge h(\texttt{id}'', y''') = h(\texttt{id}''', y'')]$ and

$$D = \mathbf{E}[s(\texttt{id}, y') s(\texttt{id}', y) s(\texttt{id}'', y''') s(\texttt{id}''', y'')]$$

Since $s$ is 4-way independent, $D$ is 1 when every pair in $\{(\texttt{id}, y'), (\texttt{id}', y), (\texttt{id}'', y'''), (\texttt{id}''', y'')\}$ appears at least twice, and 0 otherwise. So, $C \cdot D$ is $1/b^2$ when either $(\texttt{id}, y') = (\texttt{id}'', y''') \neq (\texttt{id}', y) = (\texttt{id}''', y'')$ or $(\texttt{id}, y') = (\texttt{id}''', y'') \neq (\texttt{id}', y) = (\texttt{id}'', y''')$ and 0 otherwise, so

$$\mathbf{E}[(B - \text{SumOverJoin}_{D_R, D_S}(f))^2] = \frac{1}{b^2} \sum_{\substack{(\texttt{id},x) \in D_R \\ y' \in \{1..k\} \\ (\texttt{id}',y) \in D_S}} [(\texttt{id}, y') \neq (\texttt{id'}, y)] f(x, y)^2$$

$$+ \frac{1}{b^2} \sum_{\substack{(\texttt{id},x,y) \in D_R \bowtie D_S \\ (\texttt{id}',x',y') \in D_R \bowtie D_S}} [(\texttt{id}, y') \neq (\texttt{id}', y)] f(x, y) f(x', y')$$

$$\leq \frac{1}{b^2} (k|D_R||D_S| + |D_R \bowtie D_S|^2) |f|_\infty^2 \tag{5}$$

Combining (3), (4) and (5) gives:

$$\mathrm{Var}[Q^{\mathsf{T}}C] \leq |f|_\infty^2 \left( \frac{2ke^{-\varepsilon}|D_R|}{(1-e^{-\varepsilon})^2} + \frac{1}{b^2}(k|D_R||D_S| + |D_R \bowtie D_S|^2) \right)$$

$$\leq |f|_\infty^2 |D_R|(k+1) \left( \frac{2e^{-\varepsilon}}{(1-e^{-\varepsilon})^2} + \frac{|D_S|}{b^2} \right)$$

where for the last inequality we use the fact that $|D_R \bowtie D_S| \leq \min\{|D_R|, |D_S|\}$ since by assumption identities are unique. □

## D   Optimization: Proof and generalization of Theorem 6.4 (Optimization)

Here we state and prove Thm. D.2, a generalization of Thm. 6.4. For the proof of Thm. 6.4 itself, see Section D.2.

Our general optimization theorem states that our optimization method, Algorithm 3 from §6, produces close-to-optimal results under certain conditions. Specifically, we show that the *estimated optimal function* $\hat{f}$, which we define in Definition 6.3 and show how to compute in §6.2, has a value close to the true optimum $f_{\mathrm{opt}} = \arg\min_{f \in \mathcal{F}} \mathrm{SumOverJoin}_{D_R,D_S}(f)$. Our bound on the accuracy of $\hat{f}$ given in Thm. D.2 depends on a quality score (Definition D.1) and the uniform covering number of the function class.

**Definition D.1.** The *quality rating* $\kappa(D_R, D_S, k, b, \varepsilon) > 0$ depends on a pair of datasets $D_R \subseteq \mathcal{U} \times X$ and $D_S \subseteq \mathcal{U} \times \{1..k\}$, an integer $b > 0$ (number of hash buckets) and a real number $\varepsilon > 0$ (privacy parameter), and is defined as:

$$\kappa(D_R, D_S, k, b, \varepsilon) = \frac{|D_R \bowtie D_S|W_R W_S}{k\phi + \psi}$$

where

$$\phi = \frac{|D_R|}{|D_R \bowtie D_S|} \quad \psi = \frac{|D_S|}{|D_R \bowtie D_S|} - 1$$

$$W_R = \min\left\{1, \frac{b}{5k|D_R|}\right\} \quad W_S = \min\left\{\frac{1}{4}, \varepsilon, \frac{1}{14}\sqrt{\frac{b}{|D_S|}}\right\}$$

**Theorem D.2** (Generalization of Thm. 6.4). *Fix a pair of datasets $D_R, D_S$, parameters $b$ (number of hash buckets) and $\varepsilon > 0$, a bound $M > 0$, a class $\mathcal{F}$ of functions $f : X \times Y \to [-M, M]$, and an error parameter $\mu > 0$. Let $C = \mathrm{PrivCountSketch}_{b,h,s,\varepsilon}(D_S)$ be the sketch sent by S, and*

$$f_{\mathrm{opt}} = \arg\min_{f \in \mathcal{F}} \mathrm{SumOverJoin}_{D_R,D_S}(f)$$

$$\hat{f} = \arg\min_{f \in \mathcal{F}} \Phi_{D_R,C}(f)$$

*That is, $f_{\mathrm{opt}}$ and $\hat{f}$ are the true optimal function and the function returned by Algorithm 3, respectively. Then*

$$\Pr\left[ \frac{\mathrm{SumOverJoin}_{D_R,D_S}(\hat{f})}{|D_R \bowtie D_S|} > \frac{\mathrm{SumOverJoin}_{D_R,D_S}(f_{\mathrm{opt}})}{|D_R \bowtie D_S|} + \mu \right]$$

$$\leq (c+1)\exp\left( -\Omega\left( \frac{\mu^2 \kappa(D_R, D_S, k, b, \varepsilon)}{M^2} \right) \right)$$

*where $c$ is the uniform covering number*

$$c = \mathcal{N}_\infty\left( \frac{\mu W_R W_S}{72k\phi}, \mathcal{F}, |D_R \bowtie D_S| \right)$$

*and $\kappa, \phi, W_R, W_S$ are given in Definition D.1.*

For reference, we define $\mathcal{N}_\infty$ in Appendix B.4. We prove Thm. D.2 through a careful combination of tail bounds. It has similarities to classic machine learning results relating empirical loss to expected loss on an underlying distribution, but note that we instead relate two empirical values: the sum of a function over a true join, versus what we recover from the sketch.

## D.1 Proof of Theorem D.2

Central to the proof is the following bound on the error in the score of a single function:

**Lemma D.3.** *Fix a pair of datasets $D_R, D_S$, parameters $b$ (number of hash buckets) and $\varepsilon > 0$, and a bound $M > 0$. Then there is a real number $\beta = \beta(D_R, D_S, b, \varepsilon) \geq \frac{1}{18} W_R W_S$, where $W_R$ and $W_S$ are as in Definition D.1, such that for every function $f : X \times Y \to [-M, M]$, the value $\beta^{-1} \Phi_{D_R,C}(f)$ approximates $\mathrm{SumOverJoin}_{D_R,D_S}(f)$ in the following sense.*

*Let the hash functions $h : \mathcal{U} \to \{1..b\}$ and $s : \mathcal{U} \to \{-1, 1\}$ be sampled uniformly at random, and let $C = \mathrm{PrivCountSketch}_{b,h,s,\varepsilon}(D_S)$. Then for every $\mu > 0$,*

$$\Pr\left[\frac{|\beta^{-1}\Phi_{D_R,C}(f) - \mathrm{SumOverJoin}_{D_R,D_S}(f)|}{|D_R \bowtie D_S|} \geq \mu\right]$$
$$\leq 4\exp\left(-\frac{\mu^2\,\kappa(D_R, D_S, k, b, \varepsilon)}{720 M^2}\right)$$

We first show how to prove Thm. D.2 from Lemma D.3, and then prove Lemma D.3 in §D.1.1.

*Proof of Thm. D.2 from Lemma D.3.* The proof can be summarized as a chain of two approximate equalities and one inequality:

$$\mathrm{SumOverJoin}_{D_R,D_S}(\hat{f}) \approx \beta^{-1}\Phi_{D_R,C}(\hat{f}) \leq \beta^{-1}\Phi_{D_R,C}(f_{\mathrm{opt}})$$
$$\approx \mathrm{SumOverJoin}_{D_R,D_S}(f_{\mathrm{opt}})$$

where each of the $\approx$ is proved by Lemma D.3, and the $\leq$ follows by the definition of $\hat{f}$. Unfortunately, since $\hat{f}$ depends on some randomness (the noise added for privacy, and the hash functions), we cannot directly prove the first $\approx$ through an application of Lemma D.3, and so we first find a cover of functions, and apply the Lemma to each function in the cover.

Let $J \subseteq X \times Y$ be the inner join $D_R \bowtie D_S$ excluding the id column. Let $V \subseteq \mathbf{R}^{|J|}$ be a set of size $c$ which is an $\ell_\infty \frac{\mu W_R W_S}{72k\phi}$-cover of $\mathcal{F}|_J$.

For any $f, f' \in \mathcal{F}$ satisfying the "closeness" condition guaranteed by the cover; that is, as long as

$$\forall (x, y) \in J \; |f(x, y) - f'(x, y)| < \frac{\mu}{4k\phi}$$

is true, we have

$$|\mathrm{SumOverJoin}_{D_R,D_S}(f) - \mathrm{SumOverJoin}_{D_R,D_S}(f')| \leq \sum_{(\mathtt{id},x,y)\in D_R\bowtie D_S} |f(x, y) - f'(x, y)|$$
$$\leq |D_R \bowtie D_S|\frac{\mu W_R W_S}{72k\phi}$$
$$\leq \frac{\mu}{4}|D_R \bowtie D_S| \tag{6}$$

and similarly, from Definition 6.2 and the triangle inequality, together with Lemma D.8 (which says $\beta \geq \frac{1}{18} W_R W_S$),

$$|\beta^{-1}\Phi_{D_R,C}(f) - \beta^{-1}\Phi_{D_R,C}(f')| \leq \beta^{-1}\sum_{(\mathtt{id},x)\in D_R}\sum_{y=1}^{k}\frac{1}{N_R(h(\mathtt{id}, y))}\cdot$$
$$|s(\mathtt{id}, y)\,\mathrm{clip}(C)_{h(\mathtt{id},y)}(f(x, y) - f'(x, y))|$$
$$\leq \frac{18}{W_R W_S}k|D_R|\frac{\mu W_R W_S}{72k\phi}$$
$$= \frac{\mu}{4}|D_R \bowtie D_S| \tag{7}$$

(For the last step, recall $\phi = \frac{|D_R|}{|D_R\bowtie D_S|}$.)

Let $E$ be the event that every function in $V$, as well as $f_{\mathrm{opt}}$, has its value approximated by $\beta^{-1}$ times its score within $\frac{\mu}{4}$:

$$E = \left\{ \forall f \in V \cup \{f_{\mathrm{opt}}\} \middle| \left| \frac{\mathrm{SumOverJoin}_{D_R,D_S}(f)}{|D_R \bowtie D_S|} - \frac{\beta^{-1}\,\Phi_{D_R,C}(f)}{|D_R \bowtie D_S|} \right| < \frac{\mu}{4} \right\}$$

By combining Lemma D.3 (with $\frac{\mu}{4}$ in the place of $\mu$) with a union bound over every $f \in V \cup \{f_{\mathrm{opt}}\}$,

$$\Pr[E^c] \leq 4(|V|+1)\exp\left( -\frac{\mu^2\,\kappa(D_R,D_S,k,b,\varepsilon)}{11520M^2} \right)$$

Assume $E$ holds and let $\hat{f}'$ be a function in $V$ which is close to $\hat{f}$; that is, $\forall (x,y) \in J |\hat{f}(x,y) - \hat{f}'(x,y)| < \frac{\mu}{4k\phi}$. Then, using (6) and (7), and the fact that by definition $\Phi_{D_R,C}(\hat{f}) \leq \Phi_{D_R,C}(f_{\mathrm{opt}})$:

$$\frac{\mathrm{SumOverJoin}_{D_R,D_S}(\hat{f})}{|D_R \bowtie D_S|} \leq \frac{\mathrm{SumOverJoin}_{D_R,D_S}(\hat{f}')}{|D_R \bowtie D_S|} + \frac{\mu}{4}$$

$$\leq \frac{\beta^{-1}\,\Phi_{D_R,C}(\hat{f}')}{|D_R \bowtie D_S|} + 2\frac{\mu}{4}$$

$$\leq \frac{\beta^{-1}\,\Phi_{D_R,C}(\hat{f})}{|D_R \bowtie D_S|} + 3\frac{\mu}{4}$$

$$\leq \frac{\beta^{-1}\,\Phi_{D_R,C}(f_{\mathrm{opt}})}{|D_R \bowtie D_S|} + 3\frac{\mu}{4}$$

$$\leq \frac{\mathrm{SumOverJoin}_{D_R,D_S}(f_{\mathrm{opt}})}{|D_R \bowtie D_S|} + 4\frac{\mu}{4} \qquad \square$$

#### D.1.1 Proof of Lemma D.3

Restated for convenience:

**Lemma D.3.** *Fix a pair of datasets $D_R, D_S$, parameters $b$ (number of hash buckets) and $\varepsilon > 0$, and a bound $M > 0$. Then there is a real number $\beta = \beta(D_R, D_S, b, \varepsilon) \geq \frac{1}{18}W_R W_S$, where $W_R$ and $W_S$ are as in Definition D.1, such that for every function $f : X \times Y \to [-M, M]$, the value $\beta^{-1}\,\Phi_{D_R,C}(f)$ approximates $\mathrm{SumOverJoin}_{D_R,D_S}(f)$ in the following sense.*

*Let the hash functions $h : \mathcal{U} \to \{1..b\}$ and $s : \mathcal{U} \to \{-1,1\}$ be sampled uniformly at random, and let $C = \mathrm{PrivCountSketch}_{b,h,s,\varepsilon}(D_S)$. Then for every $\mu > 0$,*

$$\Pr\left[ \frac{|\beta^{-1}\,\Phi_{D_R,C}(f) - \mathrm{SumOverJoin}_{D_R,D_S}(f)|}{|D_R \bowtie D_S|} \geq \mu \right]$$
$$\leq 4\exp\left( -\frac{\mu^2\,\kappa(D_R, D_S, k, b, \varepsilon)}{720M^2} \right)$$

To prove Lemma D.3, we begin by describing the factor $\beta$ in (Definition D.7) and bounding it (Lemma D.8). Next, we show $\Phi_{D_R,C}(f)$ has the correct value in expectation (Lemma D.9), and finally we complete the proof using a combination of tail bounds.

A technical lemma:

**Lemma D.4.** *For any two nondecreasing sequences $a_1 \geq a_2 \geq \cdots \geq a_n$ and $b_1 \geq b_2 \geq \cdots \geq b_n$,*

$$\sum_{i=1}^{n} a_i b_i \geq \frac{1}{n}\left( \sum_{i=1}^{n} a_i \right)\left( \sum_{i=1}^{n} b_i \right)$$

*Proof.* First, we rewrite each side of the inequality:

$$\sum_{i=1}^{n} a_i b_i = \frac{1}{2n}\sum_{i=1}^{n}\sum_{j=1}^{n}(a_i b_i + a_j b_j)$$

and

$$\frac{1}{n}\left(\sum_{i=1}^{n} a_i\right)\left(\sum_{i=1}^{n} b_i\right) = \frac{1}{2n}\sum_{i=1}^{n}\sum_{j=1}^{n}(a_i b_j + a_j b_i)$$

It suffices to show that each term in the first sum is greater than or equal to the corresponding term in the second sum. To see this, note that

$$(a_i b_i + a_j b_j) - (a_i b_j + a_j b_i) = (a_i - a_j)(b_i - b_j)$$

and $a_i - a_j$ and $b_i - b_j$ are either both nonnegative or both nonpositive, depending on whether $i \leq j$. □

**Lemma D.5.** *Let $X = Z + \sum_{i=1}^{n} S_i$ where $Z, S_1, \ldots, S_n$ are independent random variables, and $Z \sim \mathrm{TGeom}(e^{-\varepsilon})$ for some $\varepsilon > 0$ and each $S_i$ is drawn from the uniform distribution on $\{-1, 1\}$. Then*

$$\mathbf{E}[\mathrm{clip}(X)] = 0$$

*and*

$$\mathbf{E}[\mathrm{clip}(X + 1)] > \min\left\{\frac{\varepsilon}{3}, \frac{1}{6\lceil\sqrt{n}\rceil + 6}\right\}$$

*Proof.* The distribution of $X$ is symmetric around 0: $\Pr[X = x] = \Pr[X = -x]$ for any $x$. Thus, $\Pr[X \leq -1] = \Pr[X \geq 1]$, so $\mathbf{E}[\mathrm{clip}(X)] = \Pr[X \geq 1] - \Pr[X \leq 1] = 0$.

It remains to bound $\mathbf{E}[\mathrm{clip}(X + 1)]$, which equals $\Pr[X \geq 0] - \Pr[X \leq -2]$. Once again using the symmetry of $X$, we see $\Pr[X \leq -2] = \Pr[X \geq 2]$, and so

$$\mathbf{E}[\mathrm{clip}(X + 1)] = \Pr[X \geq 0] - \Pr[X \geq 2] = \Pr[X \in \{0, 1\}]$$

To avoid some repetition, let $T = \sum_{i=1}^{n} S_i$. Since $T$ has the same parity as $n$, we'll pay some attention to parities in what follows. Let $n = 2k + r$ where $r \in \{0, 1\}$ is $n$'s parity. Then

$$\Pr[X \in \{0, 1\}] = \sum_{z=-\infty}^{\infty} \Pr[T = 2z + r] \Pr[Z - 2z - r \in \{0, 1\}]$$

$$\geq \sum_{z=-z_0-r}^{z_0} \Pr[T = 2z + r] \Pr[Z - 2z - r \in \{0, 1\}]$$

where $z_0$ will be chosen later. For an integer $z$, let $a_z = \Pr[T = 2z + r]$ and $b_z = \Pr[Z - 2z - r \in \{0, 1\}]$, so

$$\mathbf{E}[\mathrm{clip}(X + 1)] = \Pr[X \in \{0, 1\}] \geq \sum_{z=-z_0}^{z_0} a_z b_z$$

Consider the sequence $z = 0, -1, 1, -2, 2, -3, 3, \ldots$: that is 0 and then $-i, i$ for each positive integer $i$. Both $(a_z)$ and $(b_z)$ are nondecreasing with respect to this order:

$$a_0 \geq a_{-1} \geq a_1 \geq a_{-2} \geq a_2 \geq \cdots$$

and

$$b_0 \geq b_{-1} \geq b_1 \geq b_{-2} \geq b_2 \geq \cdots$$

This nondecreasing property of $(a_z)$ follows because $(T + n)/2$ is a binomial random variable with mean $n/2$, and so its probability mass falls away monotonically from the mode(s) at $(T + n)/2 = \lfloor n/2 \rfloor, \lceil n/2 \rceil$, that is, $T = -r, r$. The nondecreasing property for $(b_z)$ follows from the fact that the probability mass of $Z$ falls away monotonically from the mode at $Z = 0$.

Using the above order, we may apply Lemma D.4 to the sequences $(a_z), (b_z)$, to get:

$$\mathbf{E}[\mathrm{clip}(X + 1)] = \sum_{z=-z_0-r}^{z_0} a_z b_z$$

$$\geq \frac{1}{2z_0 + r + 1}\left(\sum_{z=-z_0-r}^{z_0} a_z\right)\left(\sum_{z=-z_0-r}^{z_0} b_z\right)$$

$$= \frac{1}{2z_0 + r + 1} AB$$

where $A = \Pr[T \in [-2z_0 - r, 2z_0 + r]], B = \Pr[Z \in [-2z_0 - r, 2z_0 + r + 1]]$. From the probability mass function of $\mathrm{TGeom}(e^{-\varepsilon})$ we directly compute $B = 1 - e^{-(2z_0 + r + 1)\varepsilon}$. We apply Hoeffding's inequality to see

$$A > 1 - 2\exp\left(-\frac{(2z_0 + r)^2}{2n}\right)$$

Taking $z_0 = \lceil\sqrt{n}\rceil$ gives $A > 1 - 2e^{-2}$ and $B = 1 - e^{-(2\lceil\sqrt{n}\rceil + r + 1)\varepsilon}$. If $2\lceil\sqrt{n}\rceil + r + 1 \geq \varepsilon$, then $B \geq 1 - e^{-1}$, so we have

$$\mathbf{E}[\mathrm{clip}(X + 1)] > \frac{(1 - 2e^{-2})(1 - e^{-1})}{2\lceil\sqrt{n}\rceil + r + 1} \geq \frac{1}{6\lceil\sqrt{n}\rceil + 6}$$

Otherwise $(2\lceil\sqrt{n}\rceil + r + 1 < \varepsilon)$ we use the fact that for $x < 0$, $e^x < 1 + x + \frac{x^2}{2}$ to get $B > C\varepsilon - \frac{1}{2}C^2\varepsilon^2$ where $C = 2\lceil\sqrt{n}\rceil + r + 1 < \varepsilon$, and so

$$\mathbf{E}[\mathrm{clip}(X + 1)] > \frac{1}{C}(1 - 2e^{-2})(C\varepsilon - \tfrac{1}{2}C^2\varepsilon^2)$$
$$= (1 - 2e^{-2})(1 - \tfrac{1}{2}C\varepsilon)\varepsilon$$
$$> \tfrac{1}{3}\varepsilon \qquad\qquad \square$$

**Lemma D.6.** *If $X$ is a binomially-distributed random variable $X \sim \mathrm{Binom}(n, p)$ then*

$$\Pr[X \geq \max\{5np, 1\}] < \frac{1}{3}$$

*Proof.* Let $P = \Pr[X \geq \max\{5np, 1\}] = \min\{\Pr[X \geq 5np], \Pr[X \geq 1]\}$.

If $np < \frac{1}{3}$, then using Markov's inequality,

$$P \leq \Pr[X \geq 1] \leq \mathbf{E}[X] = np < \frac{1}{3}$$

Otherwise $(np \geq \frac{1}{3})$, we apply the Chernoff bound in the following form:

$$\Pr[X > (1 + \delta)\mathbf{E}[X]] < \left(\frac{e^\delta}{(1 + \delta)^{1+\delta}}\right)^{\mathbf{E}[X]}$$

taking $\delta = 4$ to get

$$P \leq \Pr[X \geq 5np] < \left(\frac{e^4}{5^5}\right)^{np} \leq \left(\frac{e^4}{5^5}\right)^{\frac{1}{3}} < \frac{1}{3} \qquad\qquad \square$$

**Definition D.7.** Given datasets $D_R, D_S$ and parameters $b, \varepsilon$, we define

$$\beta(D_R, D_S, b, \varepsilon) = \frac{\mathbf{E}[\Phi_{D_R, C}(\mathbf{1})]}{\mathrm{SumOverJoin}_{D_R, D_S}(\mathbf{1})}$$

where $\mathbf{1}$ is the constant function $\mathbf{1}(x, y) = 1$, and the expectation is taken over the random choice of hash functions $h, s$ and the noise from $\mathrm{TGeom}(e^{-\varepsilon})$ that party S added to $C$. (Note that $\mathrm{SumOverJoin}_{D_R, D_S}(\mathbf{1}) = |D_R \bowtie D_S|$.)

**Lemma D.8.** $\beta(D_R, D_S, b, \varepsilon) \geq \frac{1}{18}W_R W_S$ where $W_R$ and $W_S$ are as in Definition D.1.

*Proof.* For any $\mathtt{id} \in \mathcal{U}$ and $i \in \{1..k\}$, define

$$T(D_R, C, \mathtt{id}, i) := \frac{1}{N_R(h(\mathtt{id}, i))}s(\mathtt{id}, i)\,\mathrm{clip}(C)_{h(\mathtt{id}, i)}$$

So that

$$\beta(D_R, D_S, b, \varepsilon) = \frac{\mathbf{E}[\Phi_{D_R, C}(\mathbf{1})]}{|D_R \bowtie D_S|}$$
$$= \frac{1}{|D_R \bowtie D_S|}\sum_{(\mathtt{id}, x) \in D_R}\sum_{i=1}^{k}\mathbf{E}[T(D_R, C, \mathtt{id}, i)]$$

It suffices to show that for every $(\mathtt{id}, x) \in D_R$ and $i \in \{1..k\}$, if $(\mathtt{id}, y_i) \in D_S$, then

$$\mathbf{E}[T(D_R, C, \mathtt{id}, i)] \geq \frac{1}{18} W_R W_S \tag{8}$$

and otherwise $\mathbf{E}[T(D_R, C, \mathtt{id}, i)] = 0$.

Suppose $(\mathtt{id}, y_i) \notin D_S$. Then $s(\mathtt{id}, i)$ is independent of $C$ and $N_R(h(\mathtt{id}, i))$, and so the probability distribution of $T(D_R, C, \mathtt{id}, i)$ is symmetric about 0, so $\mathbf{E}[T(D_R, C, \mathtt{id}, i)] = 0$ as required.

Now suppose $(\mathtt{id}, y_i) \in D_S$. Let $N_R = N_R(h(\mathtt{id}, i))$ and define $N_S$ to be the number of rows $(\mathtt{id}', y_{i'}) \in D_S$ such that $h(\mathtt{id}', i') = h(\mathtt{id}, i)$. Note $N_R \geq 1$ because $(\mathtt{id}, x) \in D_R$, and $N_S \geq 1$ because $(\mathtt{id}, y_i) \in D_S$. Then

$$\mathbf{E}[T(D_R, C, \mathtt{id}, i)] = \sum_{n_R=1}^{|D_R|} \sum_{n_S=1}^{|D_S|} \Pr[N_R = n_R \wedge N_S = n_S] E_{n_R, n_S}$$

where
$$E_{n_R, n_S} = \mathbf{E}[T(D_R, C, \mathtt{id}, i) | N_R = n_R \wedge N_S = n_S]$$

Conditioned on a particular value $N_S = n_S \geq 1$, $s(\mathtt{id}, i) C_{h(\mathtt{id}, i)}$ is distributed as

$$1 + Z + \sum_{j=1}^{n_S - 1} S_j$$

where $Z \sim \mathrm{TGeom}(e^{-\varepsilon})$ is the noise party S added to $C$ for privacy, and $S_1, \ldots, S_{N_S - 1}$ are independent uniform samples from $\{-1, 1\}$, each having the value $s(\mathtt{id}, i) s(\mathtt{id}', i')$ for some other $(\mathtt{id}', i') \in D_S$ for which $h(\mathtt{id}', i') = h(\mathtt{id}, i)$. So, by Lemma D.5,

$$E_{n_R, n_S} > \frac{1}{n_R} \min\left\{ \frac{\varepsilon}{3}, \frac{1}{6\lceil \sqrt{n_S - 1} \rceil + 6} \right\} \tag{9}$$

In particular, $E_{n_R, n_S} \geq 0$, so in our effort to lower bound $\mathbf{E}[T(D_R, C, \mathtt{id}, i)]$, we are free to focus our attention on a subset of the terms:

$$\mathbf{E}[T(D_R, C, \mathtt{id}, i)] \geq \sum_{n_R=1}^{\lfloor B_R \rfloor} \sum_{n_S=1}^{\lfloor B_S \rfloor} \Pr[N_R = n_R \wedge N_S = n_S] E_{n_R, n_S}$$
$$\geq \Pr[N_R \leq B_R \wedge N_S \leq B_S] E_{\min}$$

where we choose

$$B_R = \min\left\{ |D_R|, 1 + \max\left\{ \frac{5k|D_R|}{b}, 1 \right\} \right\}$$

$$B_S = \min\left\{ |D_S|, 1 + \max\left\{ \frac{5|D_S|}{b}, 1 \right\} \right\}$$

and $E_{\min}$ is the minimum value of $E_{n_R, n_S}$ over all $n_R \leq B_R, n_S \leq B_S$.

Recall that to complete our proof it suffices to prove (8). For this, it suffices to show that

$$\Pr[N_R \leq B_R \wedge N_S \leq B_S] \geq \frac{1}{3} \tag{10}$$

and

$$E_{\min} \geq \frac{1}{6} W_R W_S \tag{11}$$

(10) follows from Lemma D.6 since $N_R - 1 \sim \mathrm{Binom}(n = k|D_R| - 1, p = \frac{1}{b})$ and $N_S - 1 \sim \mathrm{Binom}(n = |D_S| - 1, p = \frac{1}{b})$.

It remains to prove (11).

From (9) we have

$$E_{\min} \geq \frac{1}{\lfloor B_R \rfloor} \min \left\{ \frac{\varepsilon}{3}, \frac{1}{6\lceil \sqrt{\lfloor B_S \rfloor - 1} \rceil + 6} \right\}$$

By considering the cases $\frac{5k|D_R|}{b} \geq 1$, $\frac{5k|D_R|}{b} \leq 1$ separately it can be seen that $\frac{1}{\lfloor B_R \rfloor} \geq \min\{\frac{1}{2}, \frac{b}{10k|D_R|}\} = \frac{1}{2}W_R$. It remains only to be shown that

$$\min \left\{ \frac{\varepsilon}{3}, \frac{1}{6\lceil \sqrt{\lfloor B_S \rfloor - 1} \rceil + 6} \right\} \geq \frac{1}{3}W_S$$

We consider two cases. If $\frac{5|D_S|}{b} \leq 1$, then $B_S = 2$ and

$$\min \left\{ \frac{\varepsilon}{3}, \frac{1}{6\lceil \sqrt{\lfloor B_S \rfloor - 1} \rceil + 6} \right\} = \min \left\{ \frac{\varepsilon}{3}, \frac{1}{12} \right\}$$

Otherwise ($\frac{5|D_S|}{b} > 1$) we have $B_S = 1 + \frac{5|D_S|}{b} > 2$, and so

$$\min \left\{ \frac{\varepsilon}{3}, \frac{1}{6\lceil \sqrt{\lfloor B_S \rfloor - 1} \rceil + 6} \right\} \geq \min \left\{ \frac{\varepsilon}{3}, \frac{1}{6\left\lceil \sqrt{\frac{5|D_S|}{b}} \right\rceil + 6} \right\}$$

$$\geq \min \left\{ \frac{\varepsilon}{3}, \frac{1}{18\sqrt{\frac{5|D_S|}{b}}} \right\}$$

$$\geq \min \left\{ \frac{\varepsilon}{3}, \frac{1}{42}\sqrt{\frac{b}{|D_S|}} \right\}$$

In both cases, we find

$$\min \left\{ \frac{\varepsilon}{3}, \frac{1}{6\lceil \sqrt{\lfloor B_S \rfloor - 1} \rceil + 6} \right\} \geq \min \left\{ \frac{1}{12}, \frac{\varepsilon}{3}, \frac{1}{42}\sqrt{\frac{b}{|D_S|}} \right\} = \frac{1}{3}W_S$$

as required. □

**Lemma D.9.** *For any datasets $D_R, D_S$ and parameters $b, \varepsilon$, for uniformly random functions $h : \mathcal{U} \to \{1..b\}$ and $s : \mathcal{U} \to \{-1, 1\}$,*

$$\mathbf{E}[\Phi_{D_R, C}(f)] = \beta(D_R, D_S, b, \varepsilon)\, \mathrm{SumOverJoin}_{D_R, D_S}(f)$$

*where $C = \mathrm{PrivCountSketch}_{b,h,s,\varepsilon}(D_S)$.*

*Proof.*

$$\mathbf{E}[\Phi_{D_R, C}(f)] = \sum_{(\mathtt{id}, x) \in D_R} \sum_{i=1}^{k} f(x, y_i)\, \mathbf{E}\left[ \frac{1}{N_R(h(\mathtt{id}, i))} s(\mathtt{id}, i)\, \mathrm{clip}(C)_{h(\mathtt{id}, i)} \right]$$

$$= \sum_{(\mathtt{id}, x) \in D_R} \sum_{i=1}^{k} f(x, y_i)\, \mathbf{E}[T(D_R, C, \mathtt{id}, i)]$$

with $T$ as defined in the proof of Lemma D.8. As observed in that proof, when $(\mathtt{id}, y_i) \notin D_S$ we have $\mathbf{E}[T(D_R, C, \mathtt{id}, i)] = 0$, and so,

$$\mathbf{E}[\Phi_{D_R, C}(f)] = \sum_{(\mathtt{id}, x, y_i) \in D_R \bowtie D_S} f(x, y_i)\, \mathbf{E}[T(D_R, C, \mathtt{id}, i)] \tag{12}$$

By the definition of $\beta$,

$$\sum_{(\mathtt{id}, x, y_i) \in D_R \bowtie D_S} \mathbf{E}[T(D_R, C, \mathtt{id}, i)] = \beta(D_R, D_S, b, \varepsilon)|D_R \bowtie D_S| \tag{13}$$

Now, observe that $T(D_R, C, \mathtt{id}, i)$ has exactly the same distribution for every $(\mathtt{id}, x, y_i) \in D_R \bowtie D_S$: it is distributed as $\frac{1}{N_R}(1 + Z + \sum_{i=1}^{N_S} S_i)$ as described in the proof of Lemma D.8, where the random variables $N_R$ and $N_S$ are counts of hash collisions with $(\mathtt{id}, i)$. So, $\mathbf{E}[T(D_R, C, \mathtt{id}, i)]$ has the same value for any $(\mathtt{id}, x, y_i) \in D_R \bowtie D_S$. From (13) we see that value is exactly $\beta(D_R, D_S, b, \varepsilon)$, and so from (12) we have

$$\mathbf{E}[\Phi_{D_R,C}(f)] = \sum_{(\mathtt{id},x,y_i) \in D_R \bowtie D_S} f(x, y_i)\beta(D_R, D_S, b, \varepsilon)$$
$$= \beta(D_R, D_S, b, \varepsilon)\, \mathrm{SumOverJoin}_{D_R, D_S}(f) \qquad \square$$

*Proof of Lemma D.3.* We begin with an account of the randomness affecting $\Phi_{D_R,C}(f)$, as being in three parts: hash values $h(\mathtt{id}, i) \in \{1..b\}$, hash values $s(\mathtt{id}, i) \in \{-1, 1\}$, and the noise values $Z \sim \mathrm{TGeom}(e^{-\varepsilon})$ that were added by party S to each coordinate of the private count sketch $C$. We define a sequence of random variables representing each of these parts.

Let $\mathtt{id}_1, \mathtt{id}_2, \ldots, \mathtt{id}_{|D_R|} \in \mathcal{U}$ be the identities that appear in $D_R$. Let $s = |D_S| - |D_R \bowtie D_S|$ (the number of identities that appear in $D_S$ but not $D_R$) and let $(\mathtt{id}_1', y_{i_1}), \ldots, (\mathtt{id}_n', y_{i_s}) \in D_S$ be the entries in $D_S$ which do not participate in the join $D_R \bowtie D_S$. We define the following random variables:

- $H_{i,j} := h(\mathtt{id}_i, j) \sim \mathrm{Unif}(\{1..b\})$, for $i = 1, \ldots, |D_R|$ and $j \in \{1..k\}$;

- $S_{i,j} := s(\mathtt{id}_i, j) \sim \mathrm{Unif}(\{-1, 1\})$, for $i = 1, \ldots, |D_R|$ and $j \in \{1..k\}$;

- $H_i' := h(\mathtt{id}_j', i_j) \sim \mathrm{Unif}(\{1..b\})$ for $j = 1, \ldots, s$;

- $S_i' := s(\mathtt{id}_j', i_j) \sim \mathrm{Unif}(\{-1, 1\})$ for $j = 1, \ldots, s$; and

- $Z_i \sim \mathrm{TGeom}(e^{-\varepsilon})$ for $i \in \{1..b\}$.

We will complete the proof in two steps: first, we use Hoeffding's inequality to deal with the variables $Z_i$, and then we use McDiarmid's inequality to deal with the other four kinds of random variable $(H, S, H', S')$. The value $C$, defined next, connects the two steps together.

Let $A$ be the expected value of $\Phi_{D_R,C}(f)$ with respect to the random variables $Z_i$:

$$A = \mathbf{E}_{Z_1,\ldots,Z_b}[\Phi_{D_R,C}(f)]$$

Note that $A$ is a random variable still depending on the values $H_{i,j}, S_{i,j}, H_i', S_i'$, and

$$\mathbf{E}[A] = \mathbf{E}[\Phi_{D_R,C}(f)$$

We have

$$\Pr\left[\frac{|\beta^{-1}\Phi_{D_R,C}(f) - \mathrm{SumOverJoin}_{D_R,D_S}(f)|}{|D_R \bowtie D_S|} \geq \mu\right]$$
$$= \Pr[|\Phi_{D_R,C}(f) - \beta\,\mathrm{SumOverJoin}_{D_R,D_S}(f)| \geq \beta\mu|D_R \bowtie D_S|]$$
$$\leq \underbrace{\Pr[|\Phi_{D_R,C}(f) - A| \geq \tfrac{\beta\mu|D_R \bowtie D_S|}{2}]}_{(*)}$$
$$+ \underbrace{\Pr[|A - \beta\,\mathrm{SumOverJoin}_{D_R,D_S}(f)| \geq \tfrac{\beta\mu|D_R \bowtie D_S|}{2}]}_{(\#)}$$

We'll begin with $(*)$. The probability is taken over the choice of $h, s$ and $Z$, but we will look at the worst-case hash functions $h$ and $s$:

$$(*) \leq \max_{h_0, s_0} \Pr[|\Phi_{D_R,C}(f) - A| > \tfrac{\beta\mu|D_R \bowtie D_S|}{2} \mid h = h_0, s = s_0]$$

For fixed functions $h_0, s_0$, we can write

$$\Phi_{D_R,C}(f) = \sum_{a=1}^{b} X_a$$

where $X_a = 0$ when $N_R(a) = 0$ and otherwise

$$X_a = \frac{\text{clip}(C)_a}{N_R(a)} \sum_{(\text{id},x) \in D_R} \sum_{i=1}^{k} [h(\text{id}, i) = a] s(\text{id}, i) f(x, y_i)$$

Here, $[h(\text{id}, i) = a]$ is the indicator with value 1 if $h(\text{id}, i) = a$ and 0 otherwise. $X_a$ is bounded between $-M$ and $M$, since $N_R(a)$ equals the number of terms in the sum for which $[h(\text{id}, i) = a] = 1$. Therefore, changing the value of one variable $Z_a$ while leaving the others the same changes the corresponding $X_a$ by at most $2M$. Now, $\Phi_{D_R, C}(f)$ depends on at most $k|D_R|$ random variables $Z_a$, namely the ones for which at least one pair $(\text{id}, i)$ hashes to $a$ for some $\text{id} \in D_R$. We apply Hoeffding's inequality to get:

$$(*) \le 2 \exp\left(-\frac{\beta^2 \mu^2 |D_R \bowtie D_S|^2}{8k|D_R|M^2}\right)$$

To bound $(\#)$ we use McDiarmid's inequality (Thm. B.3). We will view $A$ as being a function of the random variables $(H_{i,j})_{i,j}, (S_{i,j})_{i,j}, (H'_i)_i, (S'_i)_i$, described above and call this function $r$:

$$r((H_{i,j})_{i,j}, (S_{i,j})_{i,j}, (H'_i)_i, (S'_i)_i) := A = \mathbf{E}_{Z_1,\ldots,Z_b}[\Phi_{D_R, C}(f)]$$

Changing the value of one hash value $H_{i,j}$ or $H'_i$ affects at most two of the variables $X_a$, and so changes $(\#)$ by at most $4M$. Changing the value of one hash value $S_{i,j}$ or $S'_i$ affects at most one of the variables $X_a$, and so changes $(\#)$ by at most $2M$. Summing up the squares of these difference values, we get $(4^2 + 2^2)(k|D_R| + s)M^2 = 20M^2$, so applying Thm. B.3, we have:

$$(\#) \le 2 \exp\left(-\frac{\beta^2 \mu^2 |D_R \bowtie D_S|^2}{40(k|D_R| + s)M^2}\right)$$

Combining these bounds with Lemma D.8, we have

$$(*) + (\#) \le 4 \exp\left(-\frac{\beta^2 \mu^2 |D_R \bowtie D_S|^2}{40(k|D_R| + s)M^2}\right)$$
$$\le 4 \exp\left(-\frac{W_R W_S \mu^2 |D_R \bowtie D_S|}{720(k\phi + \psi)M^2}\right)$$

as required. $\qquad\square$

## D.2 Proof of Theorem 6.4

We prove Thm. 6.4, repeated here for convenience. Recall that $\mathcal{F}$ is a set of classifiers, and for a function $f \in \mathcal{F}$, $f : X \to \{0, 1\}$, we define the corresponding error function $e_f(x, y) = |f(x) - y|$; for the full context, see §6.3.

**Theorem 6.4.** *For any datasets $D_R \subseteq \mathcal{U} \times X$, $D_S \times \mathcal{U} \times \{1..k\}$, function class $\mathcal{F}$, privacy parameter $\varepsilon > 0$, number of hash buckets $b$ and accuracy parameter $\mu > 0$, if $h$ and $s$ are drawn uniformly at random from the set of all[4] functions $\mathcal{U} \times \{1..k\} \to \{1..b\}$, the following holds. Let $f_{\text{opt}}$ be the classifier that minimizes $L_f$, and let $\hat{f}$ be the classifier corresponding to the function $e_{\hat{f}}$ output by Algorithm 3. Let $d$ be the VC dimension of $\mathcal{F}$, $n = |D_R \bowtie D_S|$, and*

$$\phi = \frac{|D_R|}{n} \quad \psi = \frac{|D_S|}{n} - 1 \quad W_R = \min\left\{1, \frac{b}{5k|D_R|}\right\} \quad W_S = \min\left\{\frac{1}{4}, \varepsilon, \frac{1}{14}\sqrt{\frac{b}{|D_S|}}\right\}$$

*Then*

$$\Pr\left[\frac{L_{\hat{f}} - L_{f_{\text{opt}}}}{n} > \mu\right] \le \exp\left(-\Omega\left(\frac{W_R W_S}{\phi + \psi}\mu^2 n - d \log n\right)\right)$$

We prove the theorem by combining Thm. D.2 with a standard result relating covering number to VC dimension (Lemma B.8), as well as Lemma B.9 to relate the VC dimension of $\mathcal{F}$ to that of the class $\mathcal{G}$ of error functions $e_f$.

---

[4]In practice, cryptographically secure hash functions should give the same guarantees.

*Proof.* Taking $M = 1$, Thm. D.2 (applied using the function class $\mathcal{G} = \{e_f \mid f \in \mathcal{F}\}$) says:

$$\Pr\left[\frac{\text{SumOverJoin}_{D_R,D_S}(e_{\hat{f}})}{n} > \frac{\text{SumOverJoin}_{D_R,D_S}(e_{f_{\text{opt}}})}{n} + \mu\right]$$

$$\leq (c+1)\exp(-\Omega(\mu^2\,\kappa(D_R, D_S, k, b, \varepsilon)))$$

Where $c$ is the uniform covering number

$$c := \mathcal{N}_\infty\left(\frac{\mu W_R W_S}{72k\phi}, \mathcal{G}, n\right)$$

For any $f \in \mathcal{F}$, we have $\text{SumOverJoin}_{D_R,D_S}(e_f) = L_f$, so we can rewrite that as:

$$\Pr\left[\frac{L_{\hat{f}} - L_{f_{\text{opt}}}}{n} > \mu\right] \leq (c+1)\exp(-\Omega(\mu^2\,\kappa(D_R, D_S, k, b, \varepsilon)))$$

All that remains is to bound the uniform covering number $c$. Note that the range of functions in $\mathcal{G}$ is $\{0, 1\}$, and the $\ell_\infty$ distance between points in $\{0,1\}^n$ is always 1. An examination of Definitions B.4 and B.5 then shows:

- If $\frac{\mu W_R W_S}{72k\phi} \geq 1$ then $c \leq 1$.

- If $\frac{\mu W_R W_S}{72k\phi} < 1$ then for any $\vec{y} \in (X \times \{0,1\})^n$ we have $\mathcal{N}(\frac{\mu W_R W_S}{72k\phi}, \vec{y}, \ell_\infty) = |\mathcal{G}_{\vec{y}}|$, and so $c = \max_{\vec{y} \in (X \times \{0,1\})^n} |\mathcal{G}_{\vec{y}}| = \Pi_{\mathcal{G}}(n)$.

So:

$$c \leq \Pi_{\mathcal{G}}(n)$$

Apply Lemma B.8.

$$\leq n^{\text{VCdim}(\mathcal{G})} + 1$$

Apply Lemma B.9 (recall $d = \text{VCdim}(\mathcal{F})$).

$$= n^d + 1$$

So:

$$\Pr\left[\frac{L_{\hat{f}} - L_{f_{\text{opt}}}}{n} > \mu\right] \leq (n^d + 2)\exp(-\Omega(\mu^2\,\kappa(D_R, D_S, k, b, \varepsilon)))$$

$$= \exp(-\Omega(\mu^2\,\kappa(D_R, D_S, k, b, \varepsilon) - d\log n)) \qquad \square$$

### D.3 Proof of Corollary 6.5

We prove Cor. 6.5, repeated here for convenience:

**Corollary 6.5.** *Assume $D_R$ and $D_S$ have the same set of identities, so the join is "perfect" in the sense that $|D_R| = |D_S| = |D_R \bowtie D_S|$. Let $n = |D_R \bowtie D_S|$ and let $d$ be the VC dimension of $\mathcal{F}$. Assume also that $b = \Omega(n)$. Let $f_{\text{opt}}$ be the classifier that minimizes $L_f$, and let $\hat{f}$ be the classifier corresponding to the function $e_{\hat{f}}$ output by Algorithm 3. Then for any $\mu > 0$,*

$$\Pr\left[\frac{L_{\hat{f}} - L_{f_{\text{opt}}}}{n} > \mu\right] \leq \exp(-\Omega(\mu^2 n\varepsilon - d\log n))$$

*Proof.* By Thm. 6.4, it suffices to show

$$\frac{W_R W_S}{\phi + \psi} = \Omega(\varepsilon) \tag{14}$$

Where $W_R, W_S, \phi, \psi$ are given in the theorem statement. Indeed, since by assumption $|D_R| = |D_S| = n$, we have

$$\phi = 1, \psi = 0$$

and since by assumption $b = \Omega(n) = \Omega(|D_R|) = \Omega(|D_S|)$, we have

$$W_R = \Omega(1), W_S = \Omega(\varepsilon)$$

and (14) follows. $\qquad \square$

# E  Training a binary classifier with a difference sketch

If R wishes to train a binary classification model to predict labels in $\{-1, 1\}$ that appear in $D_S$, the approach described in this section might give better results than the one described in §6.2. The reconstructed dataset it produces does not have weights, so will work with a broader class of optimization algorithms. Also, the lack of *negative* weights avoids a problem with unbounded loss functions: since the approach in §6.2 can produce a dataset with negative weights, if the loss function is not bounded above, then the training loss may not be bounded below, causing the optimization algorithm to diverge.

Instead of the usual private count sketch, S will send a *difference sketch* $C = \mathrm{DS}_{b,h,s,\varepsilon}(D_S)$ defined as follows to R, along with the parameters $b, h, s, \varepsilon$.

**Definition E.1.** The *difference sketch* $\mathrm{DS}_{b,h,s,\varepsilon}(D_S) \in \mathbf{Z}^b$ is only defined when $Y = \{-1, 1\}$ (so $k = 2$). It equals $\mathrm{PrivCountSketch}_{b,h,s,\varepsilon}(\mathrm{ids}(D_S, 1) - \mathrm{ids}(D_S, -1))$, where $\mathrm{ids}(D_S, y)$ is the collection of identities associated with $y$ in $D_S$. As usual, the four parameters $b, h, s, \varepsilon$ should be published along with $\mathrm{DS}_{b,h,s,\varepsilon}(D_S)$.

Based on this and $D_R$, party R builds a dataset $D = D(b, h, s, D_R, C)$ which is similar to the one described in §6.2, but does not have weights:

**Definition E.2.** To construct $D = D(b, h, s, D_R, C)$, for each element $(\mathtt{id}, x) \in D_R$, R checks whether $s(\mathtt{id})C_{h(\mathtt{id})}$ is positive, negative, or zero. If it is positive or negative, R adds the element $(x, 1)$ or $(x, -1)$ to $D$, respectively, and if it is zero, R does not add anything to $D$ (we say $(\mathtt{id}, x)$ is *dropped*).

One justification for using the difference sketch is that as $b \to \infty$ (so there are no hash collisions) and $\varepsilon \to \infty$ (so no noise is added for privacy), $D(b, h, s, D_R, C)$ approaches the true join $D_R \bowtie D_S$ (but without an $\mathtt{id}$ column). Besides avoiding negative weights, an advantage of using a difference sketch instead of an ordinary private count sketch is that each time R processes an element $(\mathtt{id}, x) \in D_R$, only one noise sample perturbs the result (the noise that was added to $C_{h(\mathtt{id})}$), whereas with an ordinary private count sketch, two noise samples would affect the answer (the noise values added to $C_{h(\mathtt{id},1)}$ and $C_{h(\mathtt{id},2)}$).

In our experiments, we found the approach from §6.2 sufficient. We leave the application of difference sketches to future work.

# F  Experiments with linear queries

We consider the question of how accurately we can estimate a joint distribution using linear queries. Suppose R has a categorical feature *race* that takes values such as white, asian-pac islander and etc., and S has label *income* $\leq$50K or $>$50K. In Table 1, we show the true counts of the actual number of people who fall into the combination of *race* and *income*, e.g., the fraction of asian pacific islanders who make less than 50K. We use the training set of 32,562 rows of the UCI Adult dataset in this experiment.

Below the true counts in the table are counts estimated by noisy sketches. The estimated count for the above example can be obtained by a linear query for which $f(x, y) = 1$ for $y = 0$ and $x =$ asian-pac islander. Recall that $y \in \{1..k\}$, and for this example $k = 2$ with $y = 0$ and $y = 1$ corresponding to $\leq$50K or $>$50K respectively.

For this experiment, we set $\varepsilon = 1$ and the sketch dimension is set to 500K. In the event a linear query produces a negative answer or 0, we change it 1 to avoid negative fractions. In order to compute the error of the distribution, we compute the average of the absolute difference between the true fraction who make less than 50K for each value of $x$ and the estimated fraction. Note that a linear query that involves a smaller number of people is more likely to misestimated, e.g., American Indian/Eskimo. Conversely, linear queries that involve a larger number of people are better estimated. We expect this outcome. Even if one party held all of the data, i.e., no joins, but we wanted a differentially private answer the $\mathrm{TGeom}(e^{-\varepsilon})$ noise drowns out the signal for a small number of people.

Next, we fix the sketch dimension to 500K and vary the differential privacy parameter $\varepsilon$ in the top panel of Fig. 7. Note how the error drops with increasing $\varepsilon$ as expected. We also repeat this

| | S:Income | R: White. | R: Asian-Pac Islander | R: Amer-Indian Eskimo | R: Other | R: Black |
|---|---|---|---|---|---|---|
| **True counts** | $\leq 50K$ | 20699 | 763 | 275 | 246 | 2737 |
| | >50K | 7117 | 276 | 36 | 25 | 387 |
| True fraction < 50K | | 0.74 | 0.73 | 0.88 | 0.91 | 0.88 |
| **Estimated counts** | $\leq 50K$ | 20307.0 | 774.0 | 307.0 | 201.0 | 2787.0 |
| | >50K | 7308.0 | 297.0 | 36.0 | 22.0 | 478.0 |
| Estimated fraction < 50K | | 0.74 | 0.72 | 0.90 | 0.90 | 0.85 |

Table 1: Estimating the joint distribution when R holds a feature "race" and S holds income >50K. S publishes a noisy sketch of their data and R downloads the noisy sketch along with the parameters $k, b, h, s, \varepsilon$. R's goal is to estimate the joint distribution, e.g., what fraction of the Asian-pacific islander population makes less than 50K. The true counts and fraction are shown in the top half of the table. The estimated counts are shown in the bottom half. The average error across all feature categories is 0.01. In this table, $\epsilon = 1$ and the sketch dimension is 500K.

experiment for other categorical variables such as *sex*, *education*, *marital-status* and etc. Note the feature with the highest average error: *native-country*. This is because there are some categories in this feature, e.g., Holland, with just a few people in the dataset.

In Fig. 7(b), we fix $\varepsilon = 1.0$ and show the effect of the sketch dimension on average error. As expected, error drops with increasing sketch dimension.

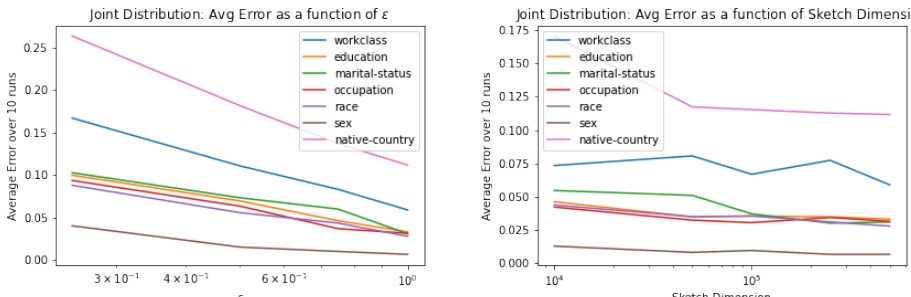

Figure 7: (a) The average error in estimating the joint distribution declines as $\varepsilon$ rises. (b) The average error fluctuates with sketch dimension less than 10K, but above 10K is relatively flat. Note that the feature *workclass* has a large number of buckets with a few people.

# G    Training details

## G.1    UCI Adult Dataset

R has the features *race, education*, etc., and S has the label *income* $\leq$50K or >50K. R would like to predict the label column given the feature columns. We one-hot encode the categorical columns for every feature and implement logistic regression in PyTorch with full batch gradient descent for 1K epochs with a learning rate of $1 \times 10^{-3}$. The dataset has 48,845 rows split into a training set of 32,562 rows and a test set of 16,283 rows. S creates a sketch for rows belonging to the train set; the test set is only used to evaluate the model.

## G.2    Extended MNIST

EMNIST is an extension of the widely used MNIST dataset of handwritten characters. We use the 240K (40K) images of digits to train (test) our protocol. The images are 28x28 pixels with labels "0" through "9". The sender S generates a private count sketch of the (id, label) pairs and the receiver R possesses the feature vectors. R would like to learn to predict labels given features.

Stochastic gradient descent plays the role of `WeightedOpt`, and the function $f(x, y)$ is modelled by a fully connected neural net with cross-entropy loss with input features $x$ and output labels $y$. The neural net has 5 hidden layers with dropout, batch norm, residual connections and ReLU

activations for a total of 1M parameters. In the non-private setting, we train it using the Adam optimizer and a learning rate of $10^{-4}$, batch size of 128 and obtain a test accuracy of 98.04%. In the presence of privacy, we conduct experiments to quantify the change in accuracy as we vary $d$, $\varepsilon$ and $|D_R|/|D_R \bowtie D_S|$. We use AdamW with a learning rate of $1 \times 10^{-5}$ and a batch size of 2048 for 200 epochs.

For the experiment involving different $k$ in Fig. 4, we use the EMNIST `bymerge` dataset of 760K images. We train a ResNet of 11M parameters, using AdamW with a batch size of 1024 and learning rate of $1 \times 10^{-6}$ for 25 epochs. For each experiment, we randomly sample $k$ classes from a total of 47 different classes, and train the model from scratch.

# H    Comparison to local differential privacy

The repository setting explored in this work is similar in some ways to *local differential privacy*, also called *randomized response* [19]. In the local model, individuals perturb their responses before releasing them, so that their privacy is preserved even if the server gathering responses is untrusted.

**Definition H.1** (Local differential privacy). Let $\varepsilon$ be a positive real number. A mechanism $M$ is $\varepsilon$-locally differentially private if it consists of applying some randomized algorithm $A$ to each element of the input and publishing the results:

$$M(x_1, \ldots, x_n) = (A(x_1), \ldots, A(x_n))$$

and that algorithm $A$ satisfies, for any two possible private values $x, x' \in X$ and any $S \subseteq \mathrm{Range}(A)$:

$$\frac{\Pr[A(x) \in S]}{\Pr[A(x') \in S]} \leq e^{\varepsilon}$$

For example, suppose we wish to estimate the fraction of people who have cancer, with $X = \{-1, +1\}$, where $+1$ means an individual has cancer. The algorithm $A$ could randomly flip an individual's answer with probability $1/(1 + e^{\varepsilon})$:

$$A(x) = \begin{cases} x, & \text{with probability } e^{\varepsilon}/(1 + e^{\varepsilon}) \\ -x, & \text{with probability } 1/(1 + e^{\varepsilon}) \end{cases}$$

and, after gathering the responses, an appropriate correction could be applied to the fraction of $+1$ responses to compensate for the flipped answers.

By contrast, in the *repository model*, illustrated in Fig. 1, each party holds a dataset of many individuals' responses keyed by unique and sensitive identifiers which must not be revealed. The challenge is for one party to release a representation of their entire dataset which is differentially private which another party can combine with their dataset to approximate the result of joining on the private identifiers. This is an instance of function release in the curator model; the definition of "differentially private" in Section 3 replaces Definition H.1.

## H.1    Is it possible to join on individual identifiers in the local setting?

Ye et al. [38] show how (key, value) pairs can be released in the local model. However, their approach only works well with keys that appear many times, so it is not a feasible way to join on identifiers which appear at most once each.

A naïve way to adapt our approach to the local model is illustrated in Fig. 8. Each individual holding a pair $(\mathtt{id}_i, y_i)$ could release a private count sketch $c_i = \mathrm{PrivCountSketch}_{b,h,s,\varepsilon}(\{(\mathtt{id}, y)\}) \in \mathbf{Z}^b$ of just their own data. The sender could then publish the sum $C = \sum_{i=1}^{n_S} c_i \in \mathbf{Z}^b$ of these individual responses, instead of $\mathrm{PrivCountSketch}_{b,h,s,\varepsilon}(D_S)$. The problem with this method is that each coordinate of $C$ has $n_S$ independent noise values $Z_\varepsilon \sim \mathrm{TGeom}(e^{-\varepsilon})$ added to it, compared to just one in the repository setting. This makes it impractical to recover any useful information from $C$.

Although this example shows the settings are different, there may be better ways to solve this problem in the local setting. Moreover, techniques from the local differential privacy literature may be useful, when appropriately adapted to the repository setting.

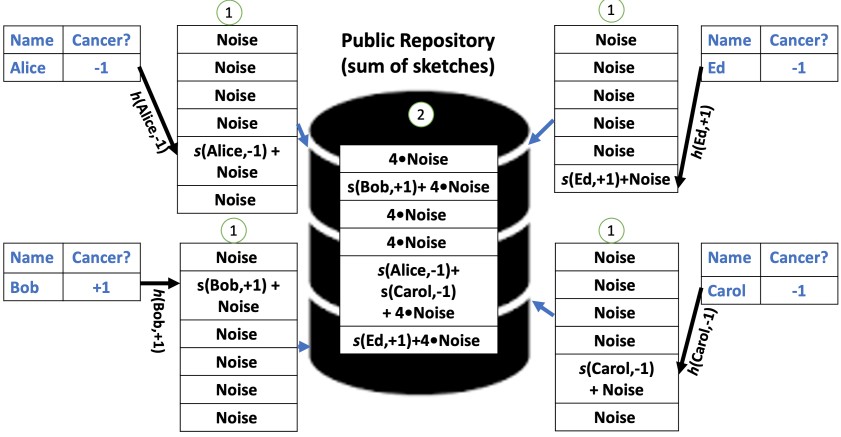

Figure 8: One interpretation of local DP in the joinable setting is that each party individually publishes a locally private sketch of their data – denoted by a "1" in a green circle. Next the public repository receives each locally private sketch and sums them up to create one sketch, denoted with "2" In the green circle. Note that the noise in the ensuing summed sketch is now a function of the number of users in the database. As the number of users in the database grows, the noise drowns out the signal. The centralized setting requires less noise. (There may be other ways of interpreting local DP in the joinable setting, this is only one interpretation.)

# I Multi-way joins and non-unique identifiers

In this appendix, we conduct some experiments to study multi-way joins and the effects of having non-unique identifiers.

## I.1 Multi-way joins

In the repository setting the receiver party can in principle join with as many senders as it wants. In §§4–6, we studied the case where the receiver joins with the dataset of one sender. Here we study the case of two senders for the Adult UCI dataset.

As before, we simulate the join setting by adding a unique *id* to each row of the dataset and split its columns. The receiver has *education*, one sender has *relationship* and the other sender has *income*. The receiver trains a logistic regression model using *education* and *relationship* as features to predict *income* (more or less than 50K). The test accuracy for different values of $\varepsilon$ is shown in blue on the right side of Fig. 9. Note that the $\varepsilon$ value is per sketch, so the total privacy cost of the three-way join (blue line) is $2\varepsilon$. For comparison, the red line shows test accuracy if the receiver does a two-way join using only the income sketch, so their model's only feature is *education*. The difference between the red and blue lines shows that for sufficiently high $\varepsilon$ the receiver is able to make use of the three-way join.

## I.2 Non-unique identifiers

We also analyse the case where the identifier is not unique. There are many practical reasons to consider such a scenario. For instance if two datasets are being joined on *legal name*, there will be several rows that correspond to different persons but have the same *legal name*. Our sketch (Definition 3.5) is well-defined even if values in the id column (*legal name* here) are repeated. However, our Theorems 5.1 and 6.4 no longer apply when identities are not unique, and it is possible to construct adversarial examples where false matches steer Algorithms 2 or 3 in the wrong direction. Finding a better solution is a great direction for future work.

In Fig. 10, we show the performance of our method on the EMNIST dataset. We first add a unique identity to the digits (240K examples), and create sender and receiver datasets with labels and images respectively. Then we add extra rows to $D_S$ which duplicate the identities of existing rows, but with

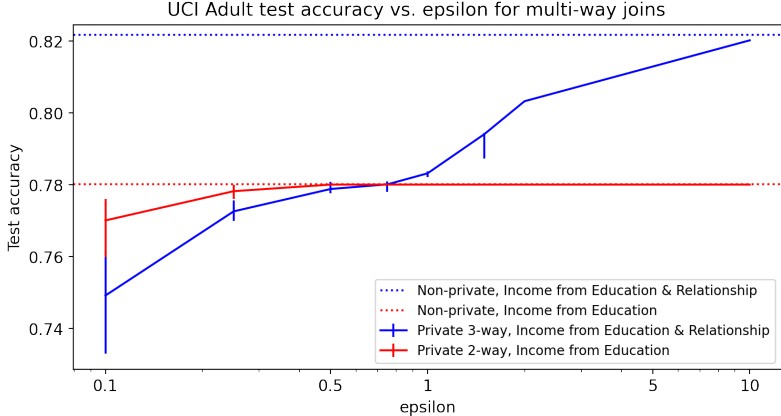

Figure 9: Test accuracy of logistic regression on the UCI Adult salary prediction task (§7.1) for a three-way join. The receiver $R$ joins his features *education*, with *relationship* and *income*. $R$ then learns a logistic regression model to predict *income* from *relationship* and *education*. The $\varepsilon$ value is per-sketch, so the total privacy budget for a 3-way join (blue line) is $2\varepsilon$.

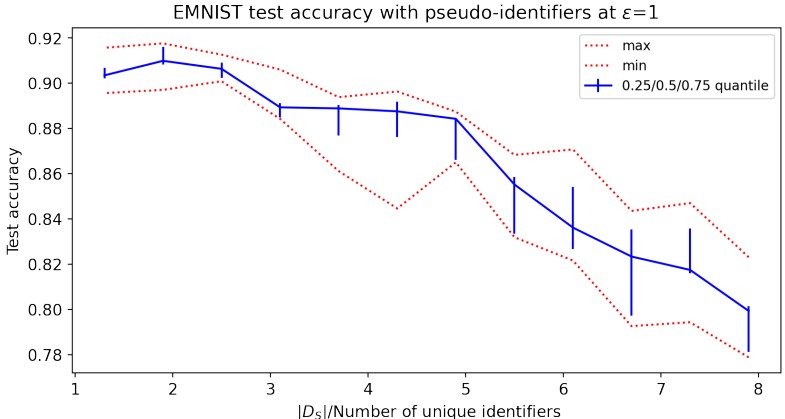

Figure 10: Test accuracy on EMNIST image classification task (§7.1) with non-unique identifiers. The $x$-axis refers to the number of times an identifier appears in $D_S$.

random labels, and apply our method with $\varepsilon = 1$. At $x = 1$ no new rows are added and at $x = 8$ each identity appears an average of 8 times in $D_S$, once with the true label and 7 times with random labels. We left $D_R$ unchanged. This simulates a situation where "pseudo" identifiers may be repeated but are independent of the data. The test accuracy on the $y$-axis shows that our method is robust and gives useful results even when the randomly-labelled false matches significantly outnumber the true matches.

Note that assuming duplicate rows have random labels is a strong assumption. Again consider the use case where *legal name* is the identifier, and the receiver has the features of a biomedical dataset. Different people with the same *legal name* might have non-trivial correlations. For instance, if a particular *legal name* is predominant amongst males, certain biological features will be correlated amongst the duplicate rows having that *legal name*. Although, it is possible that this correlation might benefit our method.

## J   Comparison with secure function evaluation

*Secure function evaluation* (SFE) protocols are cryptographic protocols for two parties holding values $x_A$ and $x_B$ to jointly compute a function $f(x_A, x_B)$ with neither party learning anything other than

the value of the function [37, 23]. If $x_A$ and $x_B$ are sets with $f(x_A, x_B) = x_A \cap x_B$ then the problem is called *private set intersection* (PSI), and if $f(x_A, x_B) = |x_A \cap x_B|$ then it is called *private set intersection cardinality* (PSI-CA). These problems can be generalized to joining datasets to answer linear queries or solve optimization problems. Unfortunately, implementations require multiple rounds of communication, and we are unaware of a protocol that would allow efficient large neural network training [35, 9, 11, 28, 27, 34]. PSI is an active area of research, and we expect future developments to increase efficiency. There are some qualitative differences between our method and PSI(-CA):

- **Interaction:** Our method requires no interaction. With PSI, two parties mutually agree to estimate the intersection of their inputs. With our method parties do not have to pre-agree on a joint function, and can use the same sketch to estimate joint distributions or train neural networks using stochastic gradient descent.

- **Exact vs. Approximate Answers:** While the generality of our sketch implies that more functions can be estimated without interaction, this flexibility comes at the cost of more noise and therefore lower accuracy than SFE, which can either give exact answers or add just enough noise to satisfy DP.

- **Simple Implementation:** SFE protocols can be quite complex to implement. This paper proposes a simple implementation with two hash functions and some noise.

- **Data Discovery in the Wild:** We are targeting a data repository of one-time sketches enabling data discovery in the wild. The proposed method lowers the barrier to private joins. Data repositories such as UCI ML repository and MIMIC (healthcare data) presently, rightly, conceal information about individuals making it challenging to join.

A sketch-based public repository enables fast and easy data discovery.

