# OpenReview forum: "Creating a Public Repository for Joining Private Data"
_NeurIPS.cc/2023/Conference — NeurIPS 2023 poster_

### Official Review · Reviewer_9EP2 · 2023-06-16

**Soundness:** 4 excellent
**Presentation:** 4 excellent
**Contribution:** 3 good
**Rating:** 7
**Confidence:** 5

**Summary:**

The paper considers a scenario where a data owner wishes to publish a private view of their data set, so that others can evaluate join (aggregations) against this data, and related questions.  Formally, the problem is to support the following problem: the data owner has a high-dimensional vector which is sparse, and wants to publish an object that will allow others to estimate the inner product of this vector with their own individual vectors.  This is represented by the SumOverJoin problem presented in Subproblem 1.2.  Building on this primitive, the paper then considers how to use this within an optimization setting, i.e., to search for a vector within some subspace that approximately optimizes some loss function.  This is sufficient to pick some approximate models, e.g. logistic regression, with a privacy guarantee.

**Strengths:**

The paper sets out a clearly defined problem, and shows how an efficient combination of sketches and noise can be applied to solve it.  The algorithm is very amenable to implementation, and performs well in experiments.

The proof that the optimization queries are accurate is quite technical and involved, demonstrating technical skill on the part of the authors.

**Weaknesses:**

The scenario may be viewed as potentially narrow: the authors describe several alternative approaches that could be used, but which fail due to the requirement that the protocol is non-interactive (e.g., private set intersection approaches).  I like the non-interactive property, but given the promotion of private set intersection protocols, it could be argued that it is not a compelling need,

It may be debatable how novel the core contribution is: there is precedent for using sketches with DP noise.  However, the extension to optimization queries goes beyond results I have seen before.

A few opportunities for improving presentation, e.g.,
"we can view the sketch as a function accepting as query an entire dataset" - I found this hard to parse
"we are not trying to compress a stream"  clarify that the objective is to take an input with n non-zeros over a domain of size D.  The goal is to be proportional to n in size, but sublinear in D.  Maybe discuss more what the solution would be if D were small.
"secure hash functions should give the same guarantees"  Perhaps more precise to say 'give the same performance'


**Questions:**

Could the optimization results be interpreted in terms of defining a subspace embedding?   I think the result follows if we can argue that the SumOverJoin bound holds for all vectors within a subspace.

**Limitations:**

No limitations in the sense of negative societal impacts.
The paper could say more about the technical limitations of the approach, e.g., extending to more complex non-linear transformations.

---

> ### Author Rebuttal · Authors · 2023-08-09
>
> > Could the optimization results be interpreted in terms of defining a subspace embedding? I think the result follows if we can argue that the SumOverJoin bound holds for all vectors within a subspace.
>
> This sounds interesting. Could you say more? Do you mean a subspace in the vector space of functions $f(x, y)$?
>
> > The paper could say more about the technical limitations of the approach, e.g., extending to more complex non-linear transformations.
>
> This is a good point. Many real-world questions can't be expressed as a linear query. We will expand our future work section.
>
> > A few opportunities for improving presentation, e.g.,  $\ldots$ Perhaps more precise to say 'give the same performance'
>
> Thank you, we will incorporate these suggestions.
>
> >It may be debatable how novel the core contribution is: there is precedent for using sketches with DP noise. However, the extension to optimization queries goes beyond results I have seen before.
>
> While there is past work on private sketches, we are not aware of past work on joinable private sketches.  Also, as you correctly point out, we are solving a more general question of joint optimization.
>
>
> ### Relation to PSI
>
> > The scenario may be viewed as potentially narrow: the authors describe several alternative approaches that could be used, but which fail due to the requirement that the protocol is non-interactive (e.g., private set intersection approaches). I like the non-interactive property, but given the promotion of private set intersection protocols, it could be argued that it is not a compelling need.
>
> Following [1], there have been many nice PSI developments over the last two decades.  These include handling malicious parties [2], ensuring all parties receive the intersection [3], joining and then aggregating over a column [4], other downstream computations on the join [5], secret sharing of the intersection [6] and more.
>
> Indeed, PSI solves a similar, but different problem. Here is a comparison:
>
> - **Interaction.** As you point out, our method requires no interaction. With PSI, two parties mutually agree to estimate the intersection of their inputs. With our method parties do not have to pre-agree on a joint function, and can use the same sketch to estimate joint distributions or train neural networks using stochastic gradient descent.
>
> - **Exact vs. Approximate Answers.** While the generality of our sketch implies that more functions can be estimated without interaction, this flexibility comes at the cost of more noise and therefore lower accuracy than SFE, which can either give exact answers or add just enough noise to satisfy DP.
>
> - **Simple Implementation.**  SFE protocols can be quite complex to implement. This submission proposes a simple implementation with two hash functions and some noise.
>
> - **Data Discovery in the Wild.** We are targeting a data repository of one-time sketches enabling data discovery in the wild.  The proposed method lowers the barrier to private joins. Data repositories such as UCI ML repository and MIMIC (healthcare data) presently, rightly, conceal information about individuals making it challenging to join.
>
>
>
> We are excited to see the types of data discoveries that will be quickly and easily possible in a sketch-based public repository. We can add an appendix providing more perspective on the strengths and weaknesses of both approaches.
>
>
>
> [1] Freedman, M.J., Nissim, K. and Pinkas, B., 2004, May. Efficient private matching and set intersection. In International conference on the theory and applications of cryptographic techniques (pp. 1-19).
>
> [2] Kissner, L. and Song, D., 2005, August. Privacy-preserving set operations. In Annual International Cryptology Conference (pp. 241-257).
>
> [3] Gordon, S.D., Hazay, C. and Le, P.H., 2022. Fully Secure PSI via MPC-in-the-Head. Proceedings on Privacy Enhancing Technologies.
>
> [4] Ion, M., Kreuter, B., Nergiz, E., Patel, S., Saxena, S., Seth, K., Shanahan, D. and Yung, M., 2017. Private intersection-sum protocol with applications to attributing aggregate ad conversions. Cryptology ePrint Archive.
>
> [5] Buddhavarapu, P., Knox, A., Mohassel, P., Sengupta, S., Taubeneck, E. and Vlaskin, V., 2020. Private matching for compute. Cryptology ePrint Archive.
>
> [6] Falk, B.H., Nema, R. and Ostrovsky, R., 2022, June. A linear-time 2-party secure merge protocol. In International Symposium on Cyber Security, Cryptology, and Machine Learning (pp. 408-427).

---

> > ### Comment · Reviewer_9EP2 · 2023-08-11
> >
> > Thank you for the detailed responses to the review comments.
> >
> > Regarding subspace embeddings, I am referring to the definition in Woodruff's survey:
> > https://arxiv.org/pdf/1411.4357.pdf
> > The point here is that the basic results for sketches consider giving a guarantee that holds for a pair of vectors (a "for each" guarantee).  Proving that the sketches form a subspace embedding means that the guarantee holds for every pair of vectors (a "for all" guarantee), which can be much stronger.
> >
> > The comparison and contrast to PSI is useful additional context, and would be nice to include in the paper.

---

> > > ### Author Response · Authors · 2023-08-20
> > >
> > > Thanks for the information about subspace embeddings. Those do seem similar in flavour to Theorem 6.4, because we do use sketching, and because we are interested in a uniform bound (over $\mathcal{F}$). We're having trouble getting the details to line up. The most obvious connection would be to make the subspace be the function class $\mathcal{F}$, since that is what we're trying to prove the uniform bound over. But $\mathcal{F}$ is not a vector subspace in general; instead we have a bound on its VC dimension.
> > >
> > > Also, it is worth noting that Theorem 6.4 isn't just about the dimensionality reduction from the count sketch. Suppose we modified our method to skip the CountSKetch, and instead send a vector indexed by every possible (identity, label) pair (an exponentially large set, or even infinite) --- in other words, make the hash function h the identity function, so no collisions are possible --- and then add noise to each entry, as before, to preserve privacy. Then Theorem 6.4 would still be nontrivial to prove, since we still need to argue that the noise does not change the result of the optimization too much.
> > >
> > > Is a connection to subspace embeddings still possible?

---

> > > > ### Comment · Reviewer_9EP2 · 2023-08-21
> > > >
> > > > Thanks for looking into this suggestion. It is worthwhile just to point out the similarities and differences with subspace embeddings, e.g., the bound on VC dimension.
> > > > Note that this is not a critical point for my evaluation -- I still endorse a positive decision for this paper regardless of this technical connection.

---

### Official Review · Reviewer_dAeb · 2023-06-30

**Soundness:** 4 excellent
**Presentation:** 4 excellent
**Contribution:** 4 excellent
**Rating:** 7
**Confidence:** 4

**Summary:**

The authors study the problem of computing a function of a join of two datasets on a user key so that the two datasets are owner by independent parties that never share their data directly. Instead the sender party gives out a DP sketch of the data that is later used by any receiver to compute an approximate answer to the query on the join. This is a very interesting formulation as it allows practical exchanges of information when the data exchanged is so sensitive (e.g., medical condition) that no party would want to participate. Contrary to other works (e.g., MPC, vertifcal Fed learning) this is one shot and does not require iterative algorithms or coordination.

The method is based on count sketch with added noise for privacy. The sender algorithm algorithm is simple to implement, efficient and elegant. The received algorithm is also straighforward for linear queries. The authors report results also on optimization of arbitrary functions. This requires that an optimizer is known for a weighted dataset ( a mild assumption). The algorithm for optimization is also very elegant requiring only a sketch-based weighting for the examples (and a factor k blow up in the number of examples to generate all possible labels)

The authors implement their method and test it for learning classification functions for up to k = 10 classes. They use standard public datasets and compare their method with a sketch-based method showing improved results.


**Strengths:**

+ the problem is very relevant to Neurips
+ the algorithm proposed are elegant and efficient.
+ theoretical analysis show utility and privacy guarantees for a wide range of applications

**Weaknesses:**

- The sender domain must be a small set {1, … k}. So this does not allow arbitrary regression (but is ok for classification in few classes).  The factor k appears in the blow up of space, time and in the utility / privacy tradeoff so it really needed to have small k.
- The comparison is only with one baseline. I agree that this area is quite new but it would be informative to compare with some other baselines with different computational models. E.g., in the case that a curator can be trusted with the label data the equivalent privacy property is Label DP. It would be interesting to see how the results here compare with the best central label DP algorithm--thus showing the price for not having a trusted curator.

Minor: /NR(h(id, y). Missing closed parenthesis.

**Questions:**

1) what is the largest k for which you find meaningful results in experiments?

2) Given the lack of baselines for this exact model, can you compare this result with the SOTA baselines in some other relaxed or more restrictive settings e.g., the trust curator model equivalent to running a central Label DP on the join, or local DP baseline (e.g., randomized response on the label)?

---

> ### Author Rebuttal · Authors · 2023-08-09
>
> Thanks for the comments and questions, based on which we did two new experiments, shown in Figure 2 of the attached pdf:
>
> ## What's the largest $k$ for which we get meaningful results? (Weakness 1, Question 1)
>
> To investigate the effect of large $k$, we used the EMNIST `bymerge` dataset consisting of 760K images of handwritten digits and uppercase and lowercase letters, with 47 classes. The left side of Figure 1 shows test accuracy as a function of the number of labels $k$.
>
> For each run, we randomly chose $k$ out of 47 classes, and applied our method with $ε=1$. The figure shows that, as expected, the performance degrades significantly as $k$ increases, but the method is still viable with $k=45$.
>
> Note that in this experiment, the size of the dataset, and thus the join size, changes as we change $k$.
>
> ## Comparing to label DP (Weakness 2, Question 2)
>
> The right side of Figure 2 compares our method to Label-DP for the EMNIST `digits` dataset of 240K handwritten digits.
>
> The definition of "neighbouring datasets" differs in our setting compared to Label-DP: we use the "add/remove" definition where a single row is added or removed; for Label-DP it is more appropriate to say a single value is changed. To account for this, we double the privacy budget for Label-DP, so for example at ε=1 on the x-axis, we actually give Label-DP a privacy budget of ε=2.
>
> The Label-DP method benefits from having a trusted curator who is able to perform the join, and so we expect it to perform better than our method, where the parties must join using a non-interactive protocol. We were surprised to see our method perform better for small ε, and do not understand why this happens.

---

> > ### Comment · Reviewer_dAeb · 2023-08-14
> >
> > Thanks for the rebuttal

---

### Official Review · Reviewer_VALz · 2023-07-06

**Soundness:** 3 good
**Presentation:** 2 fair
**Contribution:** 4 excellent
**Rating:** 7
**Confidence:** 1

**Summary:**

The paper considers the problem of publishing a privacy-preserving version of a dataset consisting of an identifier and a sensitive attribute. Additionally, it should be possible to join this published dataset by the identifiers with another dataset and compute aggregate statistics on the sensitive attributes in the joined database. This is a natural application that comes up in many scenarios, and a key innovation of this paper is the non-interactive nature of the solution. The privacy-preserving database needs to be published only once and can be reused by multiple recipients.

The approach followed is quite interesting. A dataset D \subseteq U \time {1... k} is processed as follows. For each entry (x,i) \in D store the value i + noise_i in the published table's h(x) row. In case of collisions, multiple entries are added to the same row. The noise in each row makes it differentially private.

In addition to computing aggregate statistics, the paper also considers the task of performing learning tasks over the privacy-preserving dataset.


**Strengths:**

+ The application considered is exciting.
+ The construction is relatively simple to understand.

**Weaknesses:**

- The abstract/intro does not talk about the amount of noise needed or the level of accuracy they get. It would be helpful in better understanding the results of the paper.
- The system does not define the correctness desired in this system. In particular, the joins in the proposed system will create false "positive matched rows." Does this mean that the correctness is quantified for a random receiver dataset? In particular, a particular receiver dataset can have a lot of false positive matches.


**Questions:**

- Can you please formally define the correctness and privacy properties your system satisfies?
- For my understanding, what is the magnitude of noise you expect in Z_\epsilon in Figure 5 of Appendix A for the case where the output of s is \in {0,1}?

**Limitations:**

- The paper says that the published database has DP. Can the authors expand on what this means? Typically in DP, aggregate statistics on the database are published. However, here the entire database is published. Thus, one would expect that a much larger noise parameter will be needed.
- The paper reports an increase in test results accuracy with lower noise. This makes sense but does not provide insight into the appropriate noise level.

---

> ### Author Rebuttal · Authors · 2023-08-09
>
> Thank you for these insightful comments.
>
> ## How much noise to add / how to choose ε
>
> The reviewer points out we did not clearly describe how much noise should be added. Many of the points in the review relate to this topic; we'll address them in this section.
>
> ### Specifying noise and accuracy in abstract/intro (Weakness 1, Question 2)
>
> Thanks for pointing this out. In our Contributions paragraph (end of Section 1) we will add:
>
> > We find that with a reasonable privacy parameter (ε=1) we acheive over 92% accuracy on the EMNIST dataset, and applying logistic regression on the UCI Adult dataset, we get accuracy within 1% of an algorithm trained on the original dataset with no privacy.
>
>
> $Z_ε$ in Figure 5 in the paper is sampled from TGeom($e^{-ε}$) (Def. 3.3) which for ε=1 has variance 1.84. Our method takes $s(\cdot,\cdot)\in\{-1,1\}$, but for $s(\cdot,\cdot)\in\{0,1\}$ the answer would be the same (because the sensitivity would still be 1, so we would still sample from TGeom($e^{-ε}$)).
>
> ### In the experiments, how much noise is appropriate? (Limitation 2)
>
> Our experiments show accuracy improves with less noise, but how much noise is appropriate?
>
> When applying differential privacy, one typically begins by deciding what privacy parameter ε is appropriate. This is a hard question, and the answer depends on the requirements of the party publishing the data. Ponomareva et al. give some practical advice in "How to DP-fy ML: A Practical Guide to Machine Learning with Differential Privacy". In Section 5.2.1, they suggest ε≤1 gives a strong privacy guarantee, ε≤10 gives "a reasonable level of anonymization for many applications", and ε>10 is not a sufficient guarantee of privacy.
>
> Once ε is chosen, the amount of noise necessary to achieve that guarantee is computed according to the particular method being applied. In our case, noise sampled from TGeom($e^{-ε}$) is added to each entry of the sketch (see Defs. 3.3, 3.5); this distribution has variance $2e^{-ε} / (1 - e^{-ε})^2$, which is about 1.84 when ε=1.
>
> To interpret our results, we suggest fixing ε=1 as a reasonable value; for example, when reading the right-hand side of Figs. 2 and 3, find ε=1 on the x-axis.
>
> ### Publishing a database with DP / wouldn't that require a lot of noise? (Limitation 1)
>
> The reviewer asks what it means to publish a database with DP, since typically only aggregate statistics are published.
>
> Technically, the answer comes down to the definition of differential privacy (lines 127-131 in Section 3, Preliminaries). We guarantee that for all pairs of datasets $D_1,D_2$ where $D_2$ is $D_1$ with one row added or removed, the output of our method is statistically almost the same on input $D_1$ as $D_2$, in the sense that for any set of outputs S (in our case, an output is a b-dimensional vector of integers as in Def. 3.5), the probability we output a vector in S is almost the same on input $D_1$ vs. $D_2$.
>
> As pointed out, many DP mechanisms involve publishing a few aggregate statistics with some noise added, but our method does not fall in this category. Instead, we publish a large collection of numbers, often more than there are rows in the dataset. Contrary to intuition, this does not require adding a correspondingly larger amount of noise to each of those numbers. In fact, the noise we add to each entry of the sketch depends only on the privacy parameter ε, and does not scale with the size of the sketch.
>
> The reason we can do this is that the count sketch has *sensitivity* 1. Adding an individual to the dataset adds or subtracts 1 to just one entry of the count sketch, and leaves the rest of the sketch unchanged.
>
> This reasoning applies to private count sketches in general; see e.g. "Differentially private linear sketches" by Zhao et al.
>
> Note that even though the noise we add to each number in the sketch is small, Algorithms 2 and 3 look at sums of many sketch entries, so the noise in the *outputs* of our algorithms will be large compared with methods that only look at a few aggregate statistics.
>
> ## Correctness and privacy properties (Weakness 2, Question 1)
>
> ### Correctness
>
> Thank you for pointing out that our theorem statements are missing the following information:
>
> Our theoretical guarantees (Theorems 5.1 and 6.4) apply to *any* sender and receiver datasets. We do not assume they are random.
>
> However, these theorems do assume the hash functions $h,s$ and noise values $Z$ are chosen randomly independently of the sender and receiver datasets. In particular, we do not allow the datasets to be chosen based on the particular hash functions.
>
> Because $h$ and $s$ are random, the false positive matched rows will also be random. We have designed our method so that in expectation they contribute 0 to the output of Algorithm 2, or to the score function used in Algorithm 3. This happens because for any two pairs (id,y) and (id',y'), $E[s(id,y)\cdot s(id',y')] = 0$.
>
> We will edit the theorem statements to make it clear they are quantified over all possible sender and receiver datasets, but that the randomness must be chosen independently of the datasets.
>
> ### Privacy
>
> The sketch $C$ output by Algorithm 1 is ε-differentially private. We briefly explain this in the sentence immediately after Definition 3.5, but should have made it more prominent, e.g. by marking it as a theorem or proposition or similar.

---

### Official Review · Reviewer_eoei · 2023-07-07

**Soundness:** 3 good
**Presentation:** 3 good
**Contribution:** 3 good
**Rating:** 7
**Confidence:** 2

**Summary:**

Edit: Change from Weak Accept to Accept under the expectation that the reviewer discussions are incorporated into the manuscript.

The manuscript presumes that two independent parties hold different data related to persons in a repository. As the data is vertically partitioned across the repositories, cross-referencing both repositories by joining on some person identifier yields a richer repository that interrelates more data dimensions which can be useful for data discovery/analysis.

As sharing personal data with another party could potentially lead to a violation of user privacy, the manuscript adopts the setting where party A sends a differentially private (DP) sketch of their repository to a party B such that party B can generate an approximate/noisy join result where data dimensions coming from party A can be inaccurate. The manuscript proposes a novel DP sketch to improve the accuracy of the approximate/noisy join and compares it to differentially private linear sketches (Zhao et al @ NeurIPS'22).

**Strengths:**

S1) Method proposed in manuscript satisfies pure DP unlike prior work on DP linear sketches that satisfies zCDP (which can only be translated to ADP regimes with delta > 0).

S2) Empirical results are promising: improvement over baseline and reasonable test accuracy for epsilon <= 1.

S3) Claims seem to be substantiated and key concepts are presented clearly.

**Weaknesses:**

W1) Many-to-many relationships (e.g., pseudo-identifiers) and multi-way join not explicitly considered.

W2) Not clear why the other party could not also receive a private sketch to improve the private sketch they provide to the other party (apart from complicating the setting).

W3) Minor:

p.9, l. 326: "as does private multi-dimensional sketches" => "as do private multi-dimensional sketches"

**Questions:**

Q1) How do the methods potentially generalise to pseudoidentifiers (e.g. non-unique person names) and multi-way joins with multiple private repositories (see W1).

Q2) Could a private sketch of the 2nd repository (non-private from our point of view) be helpful to guide the sketching of the private repository to improve utility when using it for join with 2nd repository? (see W2)

Q3) What does the last sentence refer to when it mentions multi-dimensional sketches? (could multiple categorical dimensions not simply be merged into one unless there are multi-dimensional range queries over numerical attributes?)

**Limitations:**

Relevant limitations seem to be discussed.

---

> ### Author Rebuttal · Authors · 2023-08-09
>
> Thank you for your insights. We have done two new experiments based on them, shown in Figure 1 in the attached pdf and described below.
>
> ## Many-to-many relationships (W1, Q1)
>
> The reviewer points out our work is limited to datasets with unique keys. In practice, we imagine repeated keys will be common for two reasons:
>
> 1. Identities may not be unique, e.g. when joining on (first name, last name).
> 2. Datasets with multiple records per person, e.g. one row for each flight a person has taken.
>
> For case 1 (non-unique IDs) one solution is to apply our methods as described. Note that the sketch (Definitions 3.3, 3.4) is well-defined even if keys are repeated. However, this approach is theoretically unsatisfactory: Theorems 5.1 and 6.4 no longer apply when identities are not unique, and it is possible to construct adversarial examples where false matches steer Algorithms 2 or 3 in the wrong direction. Finding a better solution is a great direction for future work.
>
> The left side of Figure 1 in the attached pdf shows the performance of our algorithm in a simulation of case 1. As in the paper, we added a unique identity to digits from the EMNIST dataset (240K examples), and created sender and receiver datasets $D_S, D_R$ with labels and images respectively. Then we added extra rows to $D_S$ which duplicated the identities of existing rows, but with random labels, and applied our method to $D_S, D_R$ with $ε=1$. At $x=1$ no new rows are added and at $x=8$ each identity appears an average of 8 times in $D_S$, once with the true label and 7 times with random labels. We left $D_R$ unchanged. This simulates a situation where pseudo-identifiers may be repeated but are independent of the data. The test accuracy on the y-axis shows that our method is robust and gives useful results even when the randomly-labelled false matches significantly outnumber the true matches.
>
> For case 2 (multiple records per person), in order to preserve privacy, the noise added to the private count sketch would need to scale with the maximum number of records a person can have. It would be interesting to evaluate our method empirically with this change. It should be possible to adapt Theorems 5.1 and 6.4 to this setting, probably with some loss in the bound depending on the maximum number of rows per person.
>
> Depending on the problem being solved, another approach might be to pre-process datasets to combine each identity's data into a single row. For example, an individual's history of flights could be represented just by the number of flights.
>
> We will add an appendix discussing this. A new paper (and new insights) would be needed to do the topic justice.
>
> ## Multi-way joins (W1, Q1)
>
> Joins of more than two datasets are also common in practice, and we should have addressed this.
>
> We can adapt our method to allow multiple senders' datasets. For example, looking at Figure 1 in the paper, suppose we add a second sender with age bucketed as 0-15, 16-45, 46+. Then in addition to the "Cancer +1" and "Cancer -1" columns, the receiver could add three columns corresponding to these age buckets.
>
> We tried this experimentally using three columns from the UCI Adult dataset. The receiver has *education*, one sender has *relationship* and the other sender has *income*, and we added identifiers so all three can be joined. We had the receiver train a logistic regression model using *education* and *relationship* as features to predict *income* (more or less than 50K). The test accuracy for different values of ε is shown in blue on the right side of Figure 1 in the attached pdf. Note that the ε value is per sketch, so the total privacy cost of the three-way join (blue line) is 2ε. For comparison, the red line shows test accuracy if the receiver does a two-way join using only the *income* sketch, so their model's only feature is *education*. The difference between the red and blue lines shows that for sufficiently high ε the receiver is able to make use of the three-way join.
>
> We will add an appendix.
>
> ## Using another private sketch to improve the sketch being sent (W2, Q2)
>
> Having the receiver use some hints from the sender in order to improve their sketch is a great idea. The reason we did not explore this is that we are focussed exclusively on the "non-interactive" setting, in which the sender publishes their sketch to a repository not knowing who will download it. If interaction is allowed, then existing work on cryptographic multi-party computation comes into play, and the parties are likely to get more accurate answers by using a cryptographic protocol.
>
> It would be interesting to explore a "limited-interaction" setting in which only two rounds are allowed: R sends a message to S, then S sends a message to R.
>
> ## Multi-dimensional sketches (Q3)
>
> Our future work section was too terse. By "multi-dimensional sketches" we mean sketching datasets with more than one value column --- e.g. instead of a single label in {1..k}, the sender may wish to sketch several features.
>
> It is true that multiple categorical columns could be merged into one: for example, a label in {1..3} and a label in {1..5} could be combined into a number in {1..15}. Note the performance of our method degrades as the number of possible labels increases, so this approach can only be taken so far.

---

> > ### Comment · Reviewer_eoei · 2023-08-17
> >
> > Thank you for the thoughtful and comprehensive responses!

---

### Author Rebuttal · Authors · 2023-08-09

Thanks for the thoughtful comments and suggestions. The attached figure includes experiments responding to questions from reviewers eoei and dAeb.

---

### Decision · Program_Chairs · 2023-09-21

**Decision:**

Accept (poster)

**Comment:**

This paper presents a method for producing a non-interactive differentially private sketch of a dataset that can be used to estimate joins with another dataset. The paper is well-written and provides an empirical and theoretical analysis of the algorithms.

The reviewers found this to be an interesting and somewhat surprising result. There was some discussion about how this might compare to an interactive mechanism i.e. how it compares to the baseline of running a differentially private mechanism on the two datasets together rather than separately.

Overall, the reviewers are uniformly supportive of the submission. Thus it seems appropriate for presentation at NeurIPS.